# Complex temporal dynamics of phage-bacteria populations in an animal-associated marine system

Jeffrey Liang[1,7], Karine Cahier[2,3,7], Damien Piel [2], Dario Cueva Granda[1], David Goudenège[2,3], Yannick Labreuche[2,3,6], Laurence Ma[4], Marc Monot [4], Charles Bernard[5], Eduardo P. C. Rocha [5] ✉ & Frédérique Le Roux [1] ✉

Bacteriophages-bacteria interactions drive rapid evolution of both partners in laboratory studies. To understand how these dynamics unfold in natural environments, we re-sampled a population of *Vibrio crassostreae* and their phages in an open, animal-associated marine system four years apart. Analysis of over 1000 predominantly virulent phages revealed rapid change of some lineages, but persistence of others, with genomes highly conserved between years. This pattern is consistent with low substitution rates in persistent lineages and may reflect phages overwintering in wild oysters, slow virion decay, and for temperate phages, lysogeny within hosts. Over 600 *V. crassostreae* strains recovered at both time points assorted into the same major clades. Oyster-associated vibrios have larger genomes and more abundant and diverse mobile genetic elements suggesting that oysters are hotspots for genetic exchange and horizontal gene transfer. Their genomes encode virulence plasmids, prophages carrying anti-phage systems, phage-plasmids, and phage satellites that persist intracellularly as plasmids. Time series analyses revealed weak correlations between phage and bacterial abundances, a pattern compatible with cryptic population dynamics arising from genetic diversity. Together, these results indicate that natural coevolving phage-bacteria populations can exhibit complex dynamics, with rapid replacement of some lineages alongside multi-year persistence of others.

Viruses are central players in microbial ecosystems, shaping infection dynamics, driving gene flow, and fueling antagonistic coevolution between hosts and parasites[1–4]. Bacteriophage (phage) community composition is shaped by competition among phages and antagonistic coevolution with their bacterial hosts. Each phage lineage also changes by mutation, recombination, and horizontal gene transfer (HGT). Yet

some viral populations of natural environments were found to remain genetically stable over extended periods of time for poorly understood reasons. For example, a group of *Synechococcus* cyanophages with large genomes showed remarkable genomic stability over at least 15 years in coastal waters (>99% identity in core genes)[5], even though experimental coevolution with their hosts led to rapid diversification

¹Département de microbiologie, infectiologie et immunologie & Institut Courtois d'innovation biomedicale, Université de Montréal, Montréal, QC, Canada. ²Sorbonne Université, CNRS, UMR 8227, Integrative Biology of Marine Models, Station Biologique de Roscoff, Roscoff cedex, France. ³Ifremer, Unité Physiologie Fonctionnelle des Organismes Marins, ZI de la Pointe du Diable, Plouzané, France. ⁴Institut Pasteur, Université Paris Cité, Plate-forme Technologique Biomics, Paris, France. ⁵Institut Pasteur, Université Paris Cité, CNRS UMR3525, Microbial Evolutionary Genomics, Paris, France. ⁶Present address: UMR 5244 IHPE, Université de Montpellier, CNRS, IFREMER, Université de Perpignan via Domitia, Montpellier, France. ⁷These authors contributed equally: Jeffrey Liang, Karine Cahier. ✉e-mail: eduardo.rocha@pasteur.fr; frederique.le.roux@umontreal.ca

in laboratory settings[6]. These differences highlight a central tension in viral ecology and evolution: how can viruses exhibit long-term genomic conservation in nature when they are expected to experience strong diversifying selection during antagonistic interactions with their hosts?

Ecological context—such as host density, host range, genetic diversity, and environmental conditions—may strongly influence viral evolutionary trajectories by shaping host-phage population dynamics. For example, strong phage-host interactions can generate cryptic population dynamics when rapid shifts in bacterial genotype frequencies mask changes in total host abundance, such that phage populations fluctuate while host populations appear relatively stable despite ongoing predation pressure[7]. Understanding how such eco-evolutionary processes unfold in natural systems requires longitudinal studies that integrate population structure, genetic diversity, and ecological context.

Coastal marine heterotrophic *Vibrionaceae* bacteria (hereafter vibrio) have emerged as a powerful system for addressing how phage-host interactions are structured, evolve, and persist in natural environments, supported by large-scale time-series analyses of phage-host dynamics in the wild[8]. Large-scale cross-infection studies have shown that vibrio-phage interactions often organize into modular networks structured by bacterial taxonomy and viral lineages. In parallel, work in *Vibrio lentus* populations demonstrated that phage resistance and susceptibility can evolve rapidly through the gain and loss of mobile genetic elements (MGE), resulting in quick differentiation between closely related strains[9]. Together, these studies demonstrate that phage-host coevolution in natural vibrio populations can be rapid and highly structured. Yet, they leave unresolved how such dynamics translate into patterns of persistence or turnover over longer time scales in situ.

To address this gap, we focused on an animal-associated marine system in which bacteria are exposed to a high diversity of phages. Oysters are filter feeders, continuously connected to the microbial community in the water column, yet mechanical barriers and innate immunity strongly shape the composition of their resident microbiota[10]. Within this context, we previously identified vibrio populations preferentially associated with oysters[11]. During summer mortality outbreaks, *V. crassostreae* dominates diseased juvenile oysters but is nearly absent from the surrounding seawater. The core-genome phylogeny of *V. crassostreae* revealed eight clades (V1-V8) with varying depths of divergence and with large repertoires of accessory genes[12]. Clades V2-V5 and V8 were designated phylopathotypes, as nearly all (>99%) strains in these clades carry the virulence plasmid pGV, in contrast to its lower prevalence in other clades[13,14]. Outside disease periods, virulent clades persist at low abundance in wild oysters during winter, which are far more abundant than farmed oysters and hydrodynamically connected to aquaculture sites, thereby acting as ecological reservoirs[14].

Phages infecting vibrios are also abundant and diverse in the oyster microbiota[15] where their predation may influence bacterial dynamics. In a previous time-series study, we isolated *V. crassostreae* and their phages from an oyster farm over one summer and performed exhaustive cross-infection assays[12]. This revealed a highly modular infection network, with phage adsorption tightly linked to viral genus-host clade specificity and infection further restricted by intracellular defenses encoded on MGEs. While this work provided a detailed snapshot of phage-host interactions, it raised key questions. Can the same phage genera be repeatedly isolated with the same vibrio hosts several years apart? If so, to what extent do bacteria and phage populations persist over time? Does co-occurrence within oyster shape the diversity and distribution of MGEs in *V. crassostreae* genomes?

To address these questions, we conducted a new time-series sampling four years later at the same oyster farm, generating a uniquely dense dataset for a single bacterial species-phage assemblage, comprising more than 1000 lytic phages and 600 V. *crassostreae* genomes. This allowed us to compare the bacterial and phage isolates four years apart, revealing a mixed pattern of extensive variation of some lineages and high conservation of others, despite the open marine environment. We tracked the populations of phage genera and host clades in oysters over the time series, finding temporal patterns that revealed the long-term persistence of diverse strains while rejecting simple predator-prey oscillations. Because genetic diversity can generate such dynamics, we next analyzed the repertoire of MGEs across genomes. Our results suggest that oysters act as hotspots for genetic exchanges that may fuel the complex population dynamics of phage and bacteria.

## Results

### Stable population of virulent phages within the open animal-associated marine system

To assess the temporal stability and diversification of lytic phages infecting natural vibrio populations, we resampled the same oyster farm in Brest, France, in 2021, four years after our initial survey in 2017. Across 35 dates between June 28 and September 15, we collected seawater concentrates and oyster plasma and screened them for plaque formation on 153 archival *V. crassostreae* strains representing clades V1-V8. In an initial broad screen, we combined plasma and seawater samples to maximize phage recovery for isolation and sequencing. Phages infecting clade V1 strains were consistently recovered from mid-July through nearly all subsequent sampling dates, whereas those targeting the phylopathotypes V2, V4, V5, and V8 appeared only sporadically (Fig. 1). Phages were then selected for sequencing based on host clade and sampling date, without distinguishing their habitat of origin. From 1331 isolated lytic phages, we sequenced 1033 genomes representing phages targeting all major clades of *V. crassostreae* (Supplementary Data 1). All sequenced phages belonged to *Caudoviricetes*, with genome sizes ranging from 31.6 to 187.5 kb and encoding 42 to 332 predicted genes.

Using VIRIDIC[16] we clustered the genomes into 39 genera (>70% identity; LG for Lytic Genera) and 66 species (>95%; LS for Lytic Species) (Fig. 2A and Supplementary Data 1). Using BACPHLIP[17] to computationally predict lifestyle, we classified 899 virulent phages (87% of the total) and 134 temperate phages. Consistent with the temperate lifestyle, members of LG49 and LG50 were also recovered as prophages within *V. crassostreae* genomes (see below). Each phage genus was predominantly associated with a single *V. crassostreae* clade, corresponding to the clade of the bacterial strain used for its isolation (Fig. 2A). This pattern is consistent with the clade-specific infection modules previously identified by cross-infection assays[12], although no cross-infection matrix was performed in the present study.

Among the 1033 phages sequenced in 2021, 802 (77%) were from VIRIDIC genera already isolated in 2017. All of these were predicted to be virulent. Thirteen phage species were recovered in both years, with LS29 (7 in 2017, 12 in 2021) and LS32 (28 in 2017, 259 in 2021) especially well represented—enabling direct comparisons across a four-year interval. To assess lineage persistence, we reconstructed time-calibrated phylogenies from whole-genome alignments of LS29 and LS32, correcting for recombination (Figs. 2B, C, S1–S7). Isolates from 2017 and 2021 were often very closely related, and sometimes intermingled in the tree, indicating long-term persistence of circulating lineages. LS32, a siphovirus (median genome size: 114,026 bp), showed pairwise divergence ranging from 16 to 120 single nucleotide polymorphisms (SNPs) (mean across all pairs: 70.4), with an estimated recombination-free substitution rate of $5.95 \times 10^{-5}$ substitutions per site per year. For LS29, a podovirus (52,228 bp), the rate was lower at $1.16 \times 10^{-5}$ (95% HPD: $1.55 \times 10^{-6}$-$2.33 \times 10^{-5}$), consistent with marine cyanophages and roughly tenfold lower than dairy siphoviruses[18,19]. These values fall within typical rates of substitution for dsDNA viruses

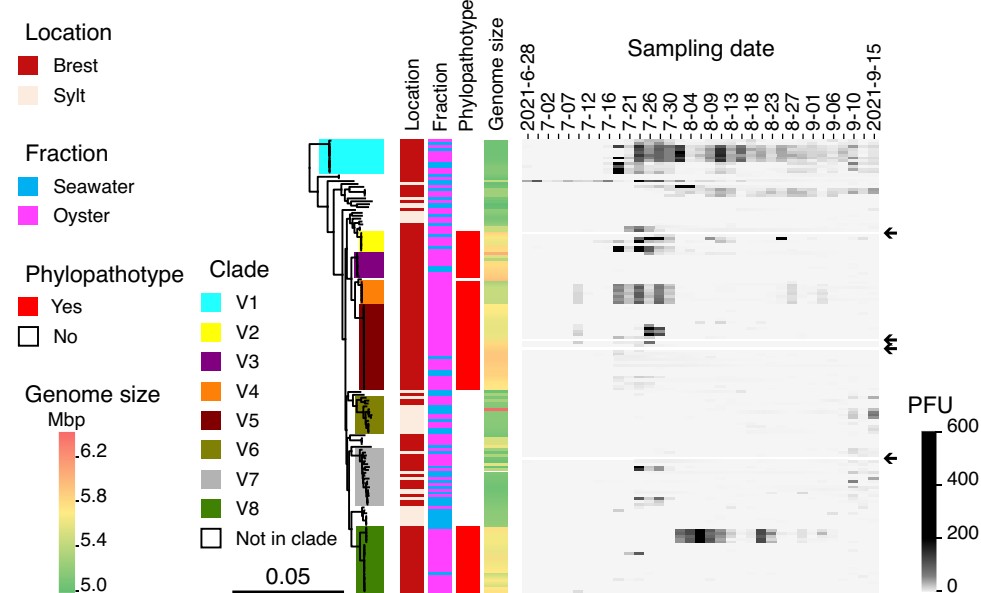

**Fig. 1 | Isolation of lytic phages infecting *Vibrio crassostreae*.** During a time-series survey in summer 2021, seawater and oyster plasma were sampled across 35 dates at a single oyster farm in the Bay of Brest, France. A panel of 153 *V. crassostreae* strains from our archival collection[12] was used as host "bait" to recover and quantify lytic phages from these environmental samples. These strains were isolated in the Bay of Brest or in Sylt (Germany) where the oyster beds have not suffered *V. crassostreae*-related disease outbreaks. Rows represent *V. crassostreae* strains ordered according to a maximum-likelihood core genome phylogeny of 157 isolates

(based on 2498 core genes), with *V. gigantis* 43_P_281 used as the outgroup to root the tree (strain not represented). Arrows indicate the 4 strains excluded due to technical issues. Columns indicate the number of plaque-forming units (PFUs) observed per strain and sampling date following infection with 10 μL of seawater viral concentrate (1000x) mixed with 10 μL oyster plasma. PFU counts are shown on a grayscale gradient. Additional metadata including clade, strain origin, habitat, genome size are provided. Source data are provided as a Source Data file.

of bacteria and eukaryotes[20]. In LS29, one 2021 isolate (D68) is identical to four isolates from 2017 and differs from the three other 2017 isolates by only 1-6 SNP (Fig. 2C), consistent with direct lineage continuity and revealing that identical phages can be isolated 4 years apart. Other 2021 isolates diverged by 14-24 SNPs, mostly in a single gene of unknown function (Fig. S8), suggestive of rapid host-driven adaptation by homologous recombination at specific genome locations. Thus, genomic analyses show that some phage lineages exhibit striking genome-wide conservation across years, with only limited and localized sequence divergence.

To evaluate whether long-term persistence of virulent phage lineages could plausibly be supported by slow particle decay, we assessed the stability of a subset of virulent phage isolates under laboratory storage conditions. High-titer stocks were stored at 4 °C and re-titered after four years. Across the phages tested, infectivity declined on average by approximately one order of magnitude (Fig. S9). Although this assay was performed on a limited number of isolates and under laboratory conditions that do not replicate environmental complexity, it indicates that some virulent *V. crassostreae* phages can remain viable over multiple years with limited loss of infectivity.

### Coexistence of phages and vibrio clades in oysters

Because phage genera are consistently associated with specific *V. crassostreae* clades, this system provides an opportunity to examine how the abundances of phages and their bacterial hosts covary across habitats and over time. To determine the habitat-specific prevalence of phages, we returned to the original samples and separately assayed plaque formation from seawater and plasma against the same hosts (Fig. 3A). This analysis revealed contrasting patterns of phage tropism. Phages infecting clade V1 were consistently recovered from both environments, suggesting a persistent environmental reservoir. In contrast, phages targeting oyster-associated phylopathotypes (V2, V3, V5, V8) were almost exclusively found in plasma, with sharp, transient

blooms spanning 9–11 days. Although plaque counts were comparable across sources, the 1000-fold concentration step required for seawater implies that actual phage densities were much higher in plasma, consistent with more intense phage-host interactions within oysters.

To validate and extend these results using a non-culture-based approach, we quantified the absolute abundance of *V. crassostreae* clades and phage genera in both habitats over time using digital droplet PCR (ddPCR). We first targeted clades V1, V2, V5, and V8 in seawater fractions and in pooled hemolymph from 90 oysters sampled across 35 time points (Fig. S10). Clade V1 was consistently detected in both habitats but remained at lower relative abundance in oysters compared to phylopathotype clades, and at similarly low levels in seawater. V1 strains were also underrepresented among cultured isolates, reflecting the greater difficulty of recovering this lineage in our collections. In contrast, clades V2 and V8, and to a lesser extent V5, exhibited pronounced temporal fluctuations—spanning up to three orders of magnitude—and reached substantially higher concentrations in oyster hemolymph than in seawater.

Quantification of phages in these samples showed that each phage genus had a higher variability than its corresponding host (Figs. S11 and S12). The seawater-associated V1-LG26 bacteria-phage pair exhibited lower coefficients of variation than the others, consistent with their relative rarity in oysters. Simple auto-correlation and partial auto-correlation analyses of both bacterial and phage time series revealed rapid temporal variation, with correlations decaying sharply beyond two sampling intervals (Fig. S13, Supplementary Data 2), suggesting largely aperiodic dynamics, as further supported by Lomb−Scargle periodogram analyses (Fig. S14). Together, these data indicate that oyster-associated phylopathotypes and their phages exhibit wide, largely non-periodic fluctuations in frequency within the host, whereas the environmental clade V1 and its associated phages persist at lower and less variable frequencies.

To examine within-host distributions of both phages and their bacterial hosts across the sampling season, we quantified abundances

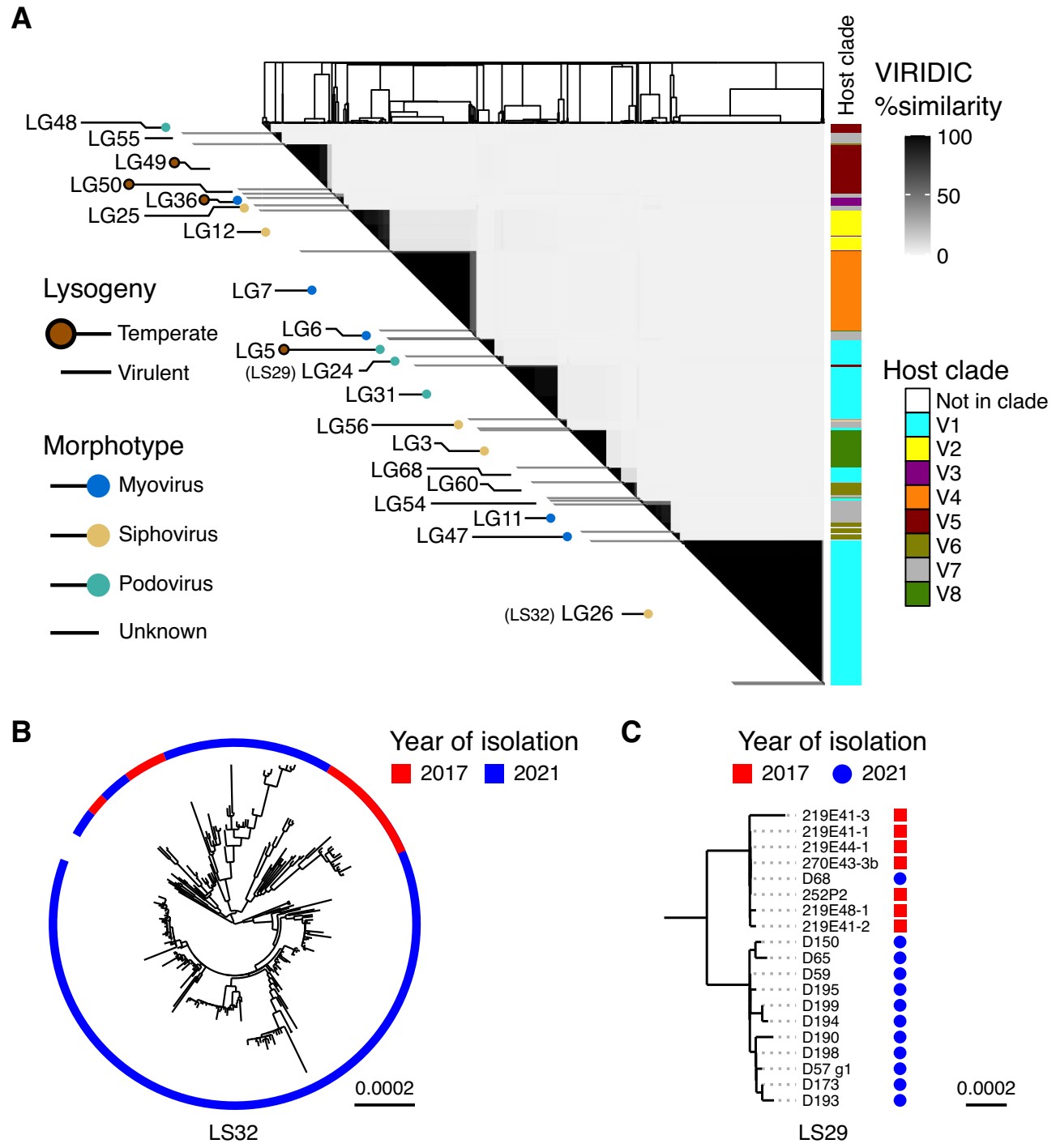

**Fig. 2 | Lytic phages form modular networks and exhibit long-term genetic stability. A** Sequenced phage genomes were hierarchically clustered based on VIRIDIC intergenomic similarities. The upper triangular matrix shows the similarity between phages with the dendrogram on the top margin showing the complete-linkage clustering. The clade of the vibrio host used to isolate each phage is annotated to the right. Phage genera with more than 3 members are annotated to the left, with red markers on the left end of each link indicating phages predicted to be temperate by BACPHLIP. Where known from transmission electron microscopy[12], the phage morphology of each genus is indicated by markers on the right end of each link. **B**, **C** Maximum-likelihood whole-genome phylogenies of lytic species of genus LG26 (including species LS32) and genus LG24 (including species LS29). Scale bars represent substitutions per site (median genome size: 144,041 bp for LS32 and 52,228 bp for LS29). In LS29, one 2021 isolate (D68) was identical to a subset of 2017 isolates, indicating clonal persistence over four years.

by ddPCR in ten oysters per time point (Fig. 3B). In some cases, peaks in phage and host clade abundance coincided. However, alpha- and beta-diversity metrics revealed strong variability among individual oysters, with no consistent temporal trends (Fig. S15A, B). The frequency of vibrio clades was positively correlated with one another within individual oysters (Fig. S15C), showing not only that each animal can simultaneously host multiple bacterial clades, but that many clades tend to co-occur. We found no clear spatial or temporal partitioning of bacterial clades or phage genera, preventing conclusions about positive associations or exclusion patterns. Correlations between vibrio clade and phage genus abundance over time were positive but low (Fig. S15C). Cross-correlation plots of vibrio and phage abundance did

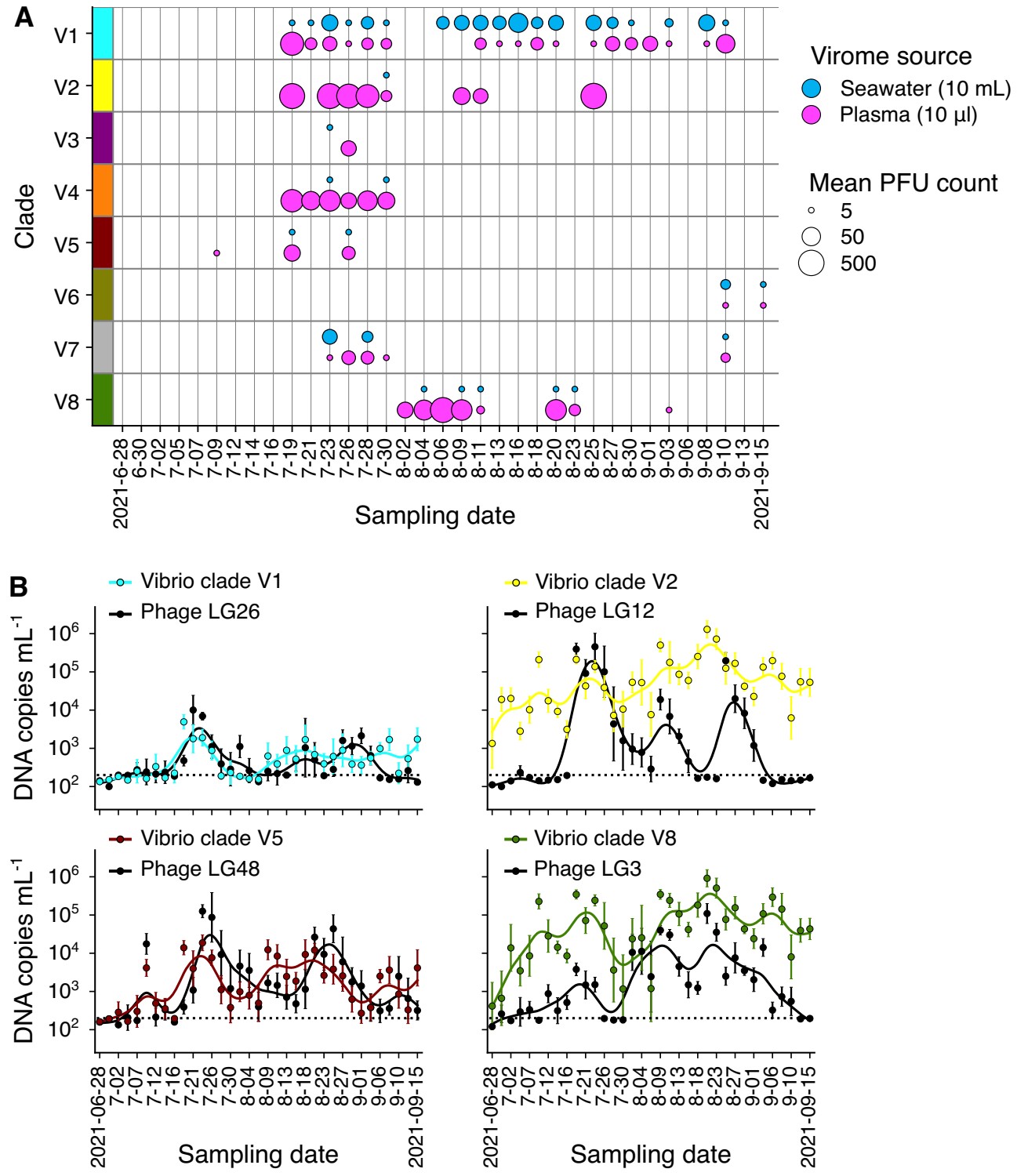

**Fig. 3 | Frequencies of phage and bacteria during the 2021 sampling season.**
**A** Quantification of the average PFU count per clade and per sampling date using 10 µL of seawater concentrate (equivalent to 10 mL of raw seawater) or 10 µL of oyster plasma pool. This approach enabled direct comparison of predation pressure across time, habitats and clades. **B** At each sampling date, 10 oysters were collected to quantify specific *V. crassostreae* clades (V1, 2, 5 and 8) and their infecting phages (respectively LG26, 12, 48 and 3) using ddPCR. Points indicate the geometric means of absolute DNA copy number per mL across the 10 oysters, with error bars showing the 95% geometric confidence intervals. For clarity, error bars are omitted where the geometric mean falls below the limit of quantification (dotted horizontal line). The overall trend in DNA quantifications is shown as an unweighted B-spline interpolated with the generalized cross validation criterion using scipy. Source data are provided as a Source Data file.

not show systematic trends suggesting strong time-lagged predator-prey oscillations (Fig. S16, Supplementary Data 3). The 4 out of 16 cases where the highest cross-correlation was observed with a non-zero-time lag all involved the phage genus LG26 or vibrio clade V1, further suggesting distinct dynamics between seawater and oyster habitats. These results show high variability in phage and vibrio populations, but little evidence of host dynamics being tightly coupled to phage abundance.

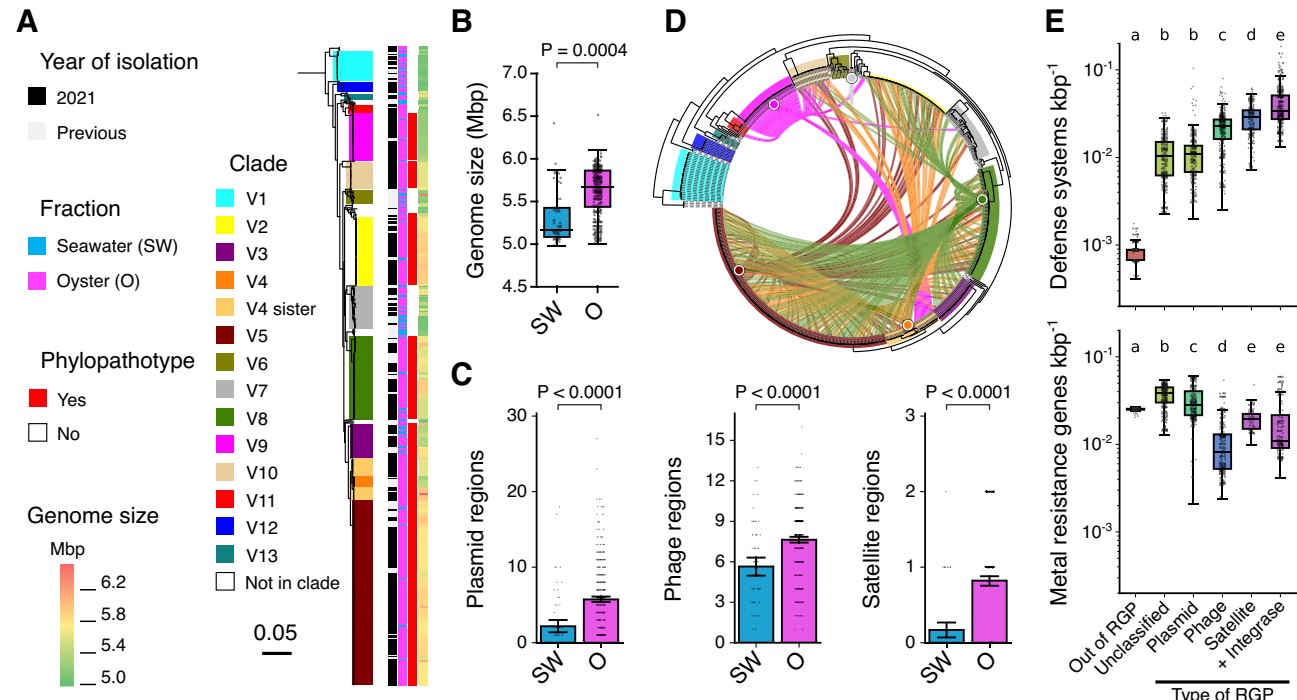

**Fig. 4 | Habitat-specific distribution of MGEs. A** Rooted maximum-likelihood phylogeny of 604 *V. crassostreae* strains based on 3099 persistent gene families (outgroup *V. gigantis* pruned). Clades and colors follow prior convention[12], with new clades (V9-V13, V4-sister) highlighted. Columns indicate year of isolation, sample type, phylopathotype status, and genome size. **B** Total genome size of assembled *V. crassostreae* genomes isolated from oysters (n = 534) and from seawater (n = 71). Data are presented as boxplots (center line, median; box, 25th–75th percentiles; whiskers, 1.5 x interquartile range) with all individual points shown. The annotation shows the two-sided significance based on 10,000 simulations of a phylogenetic ANOVA comparison between seawater and oyster genomes. **C** Counts of plasmid, temperate phage, and phage-satellite regions of genomic plasticity (RGPs) in *V. crassostreae* genomes isolated from oysters (n = 534) and from seawater (n = 71). Each bar shows the mean +/- 95% confidence interval with all individual counts shown. The annotations show the Wald test significance of a phyloGLM with Poisson_GEE regression without adjustment for multiple

comparisons to compare count data between strains (plasmid regions: $P = 2.222 \times 10^{-9}$, phage regions: $P = 4.265 \times 10^{-11}$, and satellite regions: $P = 1.281 \times 10^{-8}$). **D** Distribution of the virulence plasmid pGV across the phylogeny. Links connect strains to their closest plasmid (wGRR > 50%). **E** The coding density of phage defense systems predicted using Padloc and of metal resistance genes included in the BacMet2 database was calculated on a per kbp basis for different types of MGE and for the non-mobile genome in each strain (n = 604). Data are presented as boxplots (center line, median; box, 25th–75th percentiles; whiskers, 1.5 x interquartile range) with all individual points shown. Significant differences were confirmed using the Kruskal-Wallis H-test (defense systems: $P < 2.225 \times 10^{-308}$, metal resistance genes: $P = 3.722 \times 10^{-232}$) and group differences were calculated using Dunn's post-hoc test with Benjamini-Hochberg correction (detailed significance testing in Supplementary Data 8). Colors and letters show compact letter display groups defined by overall similarity at a significance threshold of 0.05.

## Habitat-associated vibrio genome characteristics

Having characterized the genomic and ecological dynamics of the phages, we next examined the genomic diversity of the *V. crassostreae* isolates collected during the same time series. Our goal was to determine whether the contrasting habitats experienced by oyster-associated and seawater vibrio populations are reflected in patterns of genome content and MGE diversity. We analyzed 604 sequenced isolates from previous works (157) and from 2021 (447, Supplementary Data 4). The majority of the isolates originated from oyster hemolymph (95%) and are part of the previously described clades V1-V5, V7, and V8 (78%) (Figs. 4A, S17). Many of the remaining isolates were from 5 novel clades (V9-V13), among which V9 and V10 are consistently associated with oysters.

The genome size in the vibrios range from 5.0 to 6.5 Mbp, with seawater isolates exhibiting significantly smaller genomes (n = 71; mean = 5.32 Mbp) than those from oysters (n = 534; mean genome size = 5.62 Mbp) (Fig. 4B). The phylogeny-aware analysis of gene exchange rates in the *V. crassostreae* pangenome revealed a species-wide association between core-genome phylogenetic distance and the rates of gene gain/loss ($P = 3.624 \times 10^{-051}$; Supplementary Data 5 and 6). However, this was not the case in V10 (P = 1.000) or in the seawater-associated clades V1 (P = 1.000) and V6 (P = 1.000). This is likely the result of the very recent epidemic expansion of clade V10. Clades V1 and V6 are more frequently isolated from seawater and may have

different pangenome dynamics. Accordingly, seawater isolates had significantly lower gene turnover than oyster-derived strains (P = 0.047), suggesting reduced HGT in the open-water environment.

We identified 32,349 regions of genome plasticity (RGPs) (Supplementary Data 7), comprising 3217 plasmid-associated, 4478 prophage-associated, 450 satellite-associated, and 1945 integrase-associated regions. Plasmids, prophages, and satellites (MGEs that hijack helper phages for propagation) were consistently more abundant in oyster-derived strains (Fig. 4C, Supplementary Data 8). This pattern was particularly striking for the virulence plasmid pGV, which formed a highly reticulated similarity network, indicative of frequent transfer across the species—likely mediated by the oyster-associated phylopathotypes (Fig. 4D). We next examined the associations between different MGEs and their genetic cargo. We previously showed that genes encoding anti-phage systems are enriched in RGPs relative to the core genome[12]. Here, we find that these systems are concentrated in phage-, satellite-, and integrase-associated RGPs, but comparatively rare in plasmid-associated regions (Fig. 4E, Supplementary Data 7 and 9). To further characterize plasmid diversity and function, we fully assembled 20 plasmids grouped into five families: the virulence plasmid pGV and four plasmids—p1, pMintaka, pAlioth, and pMizar (Supplementary Data 10-14). They all encoded conjugation machinery, including Type IV secretion systems and MOB relaxases, consistent with self-transmissibility. Beyond phage defense, plasmids

carry functions likely advantageous to colonize the oyster host. For instance, pGV encodes a Type VI secretion system linked to hemocyte toxicity[14] and genes potentially involved in copper resistance; p1 harbors a metallophore biosynthesis and transport system; and pMintaka encodes microcin synthesis and secretion genes. Compared to other MGEs, plasmids were enriched in functions related to oyster adaptation, including metal and biocide resistance (Fig. 4E, Supplementary Data 7)—traits likely reflecting selective pressures imposed by hemocyte activity, antimicrobial peptides, and the oyster's tight regulation of metal ions[21].

## Clade-specificity and episomal persistence of temperate phages

To assess the contribution of temperate phages to the genome plasticity of *V. crassostreae*, we screened the bacterial isolates with geNomad[22] and identified in silico ~1600 putative prophages spanning three viral classes: *Caudoviricetes* (tailed), *Tectiliviricetes* (tailless), and *Faserviricetes* (filamentous) (Supplementary Data 15). Controls using long-read sequenced genomes revealed that predictions of *Caudoviricetes* were more consistent (Fig. S18), as previously observed[23]. We thus retained 562 *Caudoviricetes* prophages larger than 25kbp, of which 525 were chromosomally integrated and 37 (6.6%) were extrachromosomal (Supplementary Data 16). VIRIDIC clustering grouped these into 36 genera (TG for temperate genus) and 80 species (TS for temperate species) (Fig. 5A). Among them, 36 species were singletons and 12 occurred in hosts not assigned to a clade. Of the remaining 32 phage species, 30 were clade-specific and 14 were found within the genomes of 2-5 vibrio clades. At the genus level, only 7 of 23 non-singleton genera (30.4%) were clade-specific—substantially fewer than for virulent phage genera (21/30, 70%). Thus, bacterial clade specificity is evident at the phage species but not genus level (Fig. 5A). We found 20 prophage species persisting in vibrio genomes sampled between 2001 and 2021, eight of which are identical across all aligned isolates. We could thus compute the substitution rates for the remaining 12 prophages, like we did for the virulent phages, dated by the isolation date of their hosts. We observed substitution rates significantly lower for prophages than for virulent phages (mean ~$10^{-6}$ substitutions per site per year) (Fig. 5B). The phylogenies of the best-represented species (Figs. S19–30) reveal striking similarities across isolates. For example, in the species TS30 more than 20 elements are identical (Fig. 5C). We further tested whether persistent phage and prophage lineages show signatures of differential selection on genes involved in phage-host interactions. We found an overall pattern of purifying selection, as dN/dS was significantly less than one for the structural and non-structural genes in both lytic phages and prophages (Fig. S31, one-sample one-sided Wilcoxon signed rank tests). These dN/dS analyses of single-copy orthologs revealed no significant differences in the distribution of predicted structural and non-structural genes (Kolmogorov-Smirnov two-sample tests-lytic: $P = 0.781$, prophage: $P = 1.0$) (Fig. S31, Supplementary Data 17). Together, these results point to tighter coevolution between vibrio clades and temperate phage species, which show limited diversity and appear to persist for years primarily through vertical transmission within clades.

The two temperate genera (TG34, TG89) correspond to LG49 and LG50, originally identified as lytic phages and then predicted temperate by BACPHLIP. Unlike most temperate elements, these stood out because we recovered them as infectious particles in plaque assays, confirming their capacity for lytic activity. They are also unusual in assembling as complete extrachromosomal contigs rather than integrated prophages. Read coverage profiles and ddPCR estimates—showing plasmid-to-chromosome ratios ranging from 3 to 10 depending on the strain—indicate a plasmid-like, low-copy-number replication strategy (Fig. S32). Both carry plasmid partition systems (Fig. 5D), supporting their identification as active phage-plasmids[24,25]. TG89 is a linear phage-plasmid, maintained extrachromosomally with covalently closed ends, resembling N15 in *E. coli*, VP882 in *V.*

*parahaemolyticus*, and phage 63 in *V. cyclitrophicus*[26,27]. It encodes *telN* (protelomerase) and *repA* (replication initiator), hallmark genes of this phage-plasmid lifestyle. TG89 also carries homologs of a small ORF and transcription factor (smORF and TF in Fig. 5D), previously implicated in polylysogeny regulation[27] though their functional roles in *V. crassostreae* are unknown.

We tracked TG34 and TG89 during the 2021 time series using ddPCR on plasma (free viral particles) and hemolymph (bacteria + viruses). TG89 remained consistently near or below the limit of quantification across all samples, precluding meaningful ecological interpretation. By contrast, TG34 exhibited sporadic but massive blooms—up to three orders of magnitude increases—in hemolymph, particularly between August 16 and August 27, during which it remained at much lower abundance in plasma (Fig. S33A). TG34 was the only phage significantly more abundant in hemolymph than plasma. Because hemolymph contains hemocytes together with associated bacteria and viruses, whereas plasma represents the cell-free fraction, this pattern is consistent with rapid reinfection and establishment as lysogens rather than long-term persistence in particle form (Fig. S33B). These phage-plasmids illustrate an additional dimension of temperate phage ecology: unlike integrated prophages, they persist as an episome and can toggle between lytic activity and lysogeny, highlighting oysters as key environments where temperate phages actively shape both viral and bacterial populations.

Taken together, our results reveal two distinct strategies of temperate phage persistence in the oyster-vibrio system: long-term vertical transmission of clade-specific prophages and episomal maintenance of phage-plasmids capable of switching between lysogeny and lysis.

## Discovery of episomal phage satellites expands the known diversity of MGEs

Two putative prophage genera, TG54 and TG93, assemble as extrachromosomal circular elements with unusually small genome sizes (15.8 and 11.1 kbp) and lack tail genes—suggesting they are phage satellites[28]. Both encode plasmid maintenance functions, including *parA*, *repA*, and a putative *ProQ/FinO*-like regulator, supporting episomal replication (Fig. 6).

TG54 was found in 30 *V. crassostreae* genomes, mostly in the oyster-associated clade V2, were isolated both in 2017 and 2021. It was classified by SatelliteFinder[29] as a capsid-forming phage-inducible chromosomal island (cf-PICI)[30], encoding homologs of structural proteins for head formation (capsid, portal, head maturation protease, head-tail adapter, closure), the HNH endonuclease and terminases for genome packaging, and an Stl-like regulator likely controlling induction in response to phage activity. TG54 lacks an integrase but encodes a distinct tyrosine recombinase located between the portal and head-tail adapter genes (Fig. 6A). The phylogeny of the capsid protein reveals clustering of TG54 with known Gammaproteobacterial cf-PICIs (Fig. 6B), clearly separated from phage capsid proteins, supporting its classification as a phage satellite rather than a cryptic phage. We propose that TG54 represents a satellite-plasmid: an element maintained vertically as a plasmid but transmitted horizontally via satellite-encoded capsids that associate with helper-phage tails to form chimeric infectious particles. Because it is not a chromosomal island, we replaced "chromosomal island" in the name of cf-PICI with "satellite-plasmid", naming this element a cf-PISP (capsid-forming phage-inducible satellite-plasmid). A search of public databases using its capsid and ParA protein sequences identified similar elements in at least two other *Vibrio* species, but none outside the genus (Fig. S34).

TG93, detected in two strains from clades V4 and V8 in 2021, carries fewer genes than cf-PISPs and was not flagged as a satellite by SatelliteFinder. Manual inspection, however, revealed divergent homologs of hallmark genes from phage-inducible chromosomal islands (PICIs), another class of phage satellites[31] (Fig. 6C). The element

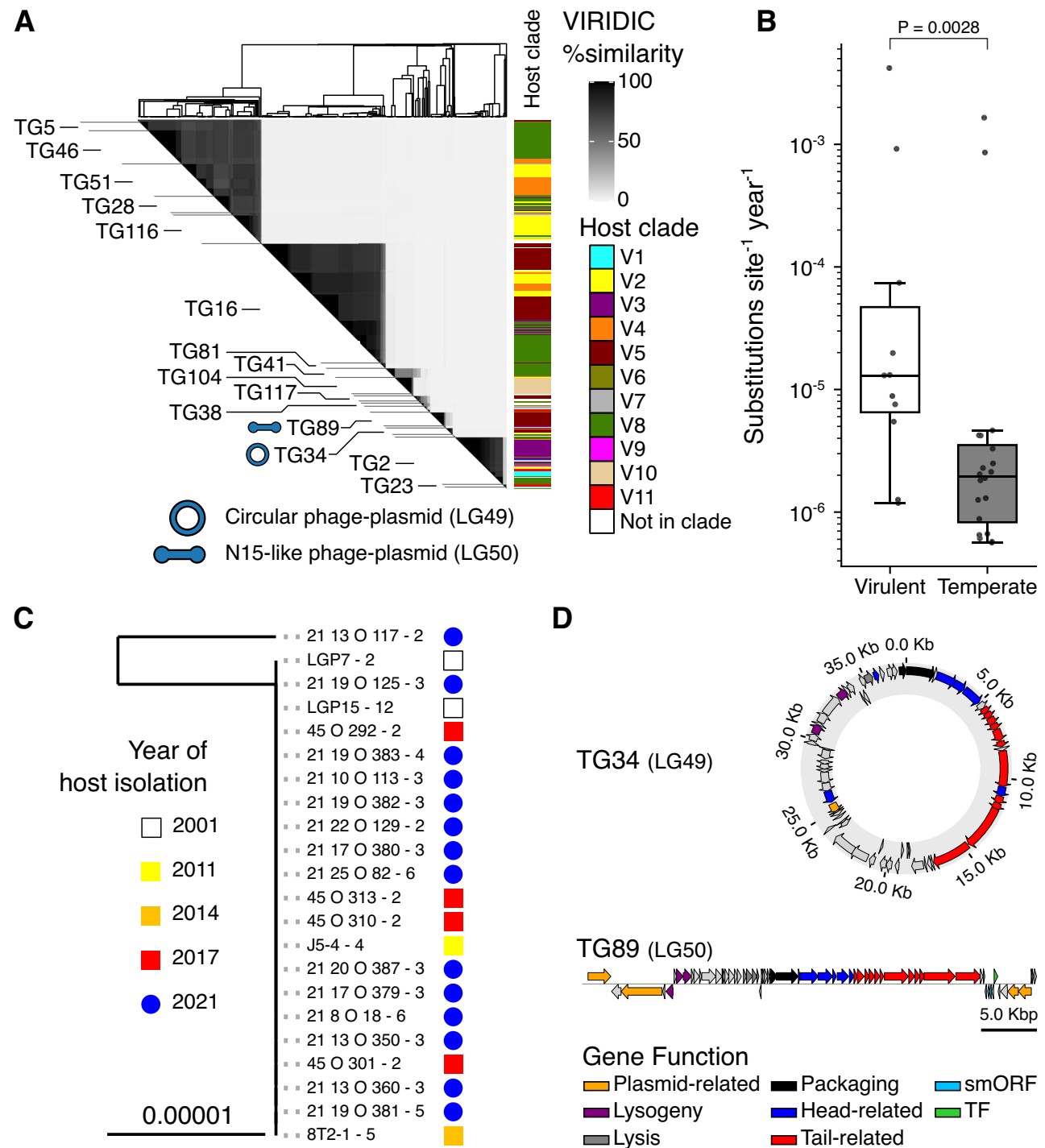

**Fig. 5 | Clade specificity and persistence of temperate phages. A** *Caudoviricete* prophages predicted by geNomad in assembled *V. crassostreae* genomes were hierarchically clustered based on VIRIDIC intergenomic similarities. The upper triangular matrix shows the similarity between prophages with the dendrogram on the top margin showing the complete-linkage clustering. The clade of the host of each prophage is annotated to the right. Prophage genera with more than 3 members are annotated to the left, with icons showing the two temperate genera also recovered as active phages in hemolymph (circular phage-plasmid LG49/TG34 and linear phage-plasmid LG50/TG89). **B** Substitution rates estimated from whole-genome alignments, calibrated with isolation dates (virus for virulent phages

($n=11$); host for prophages ($n=20$), using BEAST v2.6.3 and an exponential relaxed clock. Data are presented as boxplots (center line, median; box, 25th–75th percentiles; whiskers, 1.5 x interquartile range) with all individual points shown. Significance was assessed with a non-parametric two-sided Brunner-Munzel test. **C** The maximum-likelihood whole genome phylogeny of the TS30 prophage species was calculated with the nucleotide alignment of predicted prophage sequences extracted from the assemblies of their host bacteria. **D** TG34 (circular) and TG89 (linear) are phage-plasmids. Gene annotation revealed both plasmid- and phage-related functions. TG89 also encodes homologs of a transcription factor (TF) and smORF previously described in the linear phage-plasmid 63 of *V. cyclitrophicus*[26].

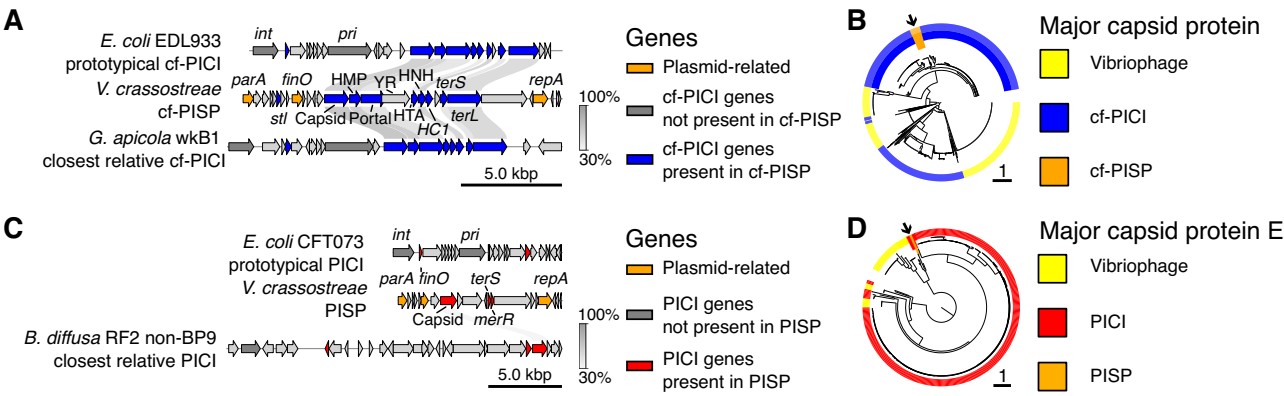

**Fig. 6 | Gene content and phylogenetic placement of satellite−plasmid elements. A** Comparison of gene content between two known capsid-forming PICIs (cf-PICIs) and the episomal element cf-PISP, showing conserved gene order. **B** Maximum-likelihood phylogeny of capsid proteins (Pfam PF05065; *n* = 1375) from cf-PISP (*n* = 30), published cf-PICIs (*n* = 900), and lytic phages from this study (*n* = 465). The arrow highlights the location of the cf-PISP subtree nested within other satellite sequences. **C** Gene content comparison between two known PICIs and the episomal element PISP, showing only weak similarity between the PISP capsid morphogenesis gene sequence and those in other PICI phage-satellites. Other labeled genes putatively encode functions analogous to those encoded in PICIs but are not detected by SatelliteFinder gene models. **D** Maximum-likelihood phylogeny of PF03864-type capsid proteins (*n* = 410) from PISP (*n* = 2), published PICIs (*n* = 356), and vibriophages from this study (*n* = 52). The arrow highlights the location of the PISP subtree nested within other satellite sequences. In (**A**) and (**C**) links indicate >30% amino acid identity based on bidirectional best hits (identified using MMseqs2). Genes are colored by functional category and presence in plasmids, phage-satellites, and/or satellite-plasmids. Tree inference was performed with IQtree with Q.pfam+F + I + R substitution model using multiple sequence alignments of proteins obtained using MAFFT L-INS-i.

encodes Capsid protein E and a head maturation protease, together with a small terminase and MerR-like transcriptional regulator. We did not detect an integrase in TG93; the gene may be dispensable in this element because it is a plasmid. Based on its gene content, we suggest that this element shares the PICI strategy of head assembly disruption, interfering with capsid subunit assembly to form a physically smaller prohead that excludes the larger helper phage genome and preferentially packages the satellite genome. The capsid protein E sequence clusters with satellite-encoded and not phage-encoded capsid proteins in the phylogeny (Fig. 6D), supporting our identification of this element as a second type of episomal satellite-plasmid. By analogy with cf-PISP, we name this smaller element PISP (phage-inducible satellite-plasmid). Related elements were detected in three other *Vibrio* species but not in any other bacterial genera (Fig. S34).

## Discussion

The study of our oyster-vibrio-phage-satellite assemblage, using over 1000 phages, 604 host genomes, and two decades of sampling at the same oyster farm, resulted in four main findings: (i) while there are many differences between the two time series, both virulent and temperate phage lineages may persist over multiple years with little genomic divergence in an open marine environment; (ii) phage and bacterial populations coexist dynamically within oysters; (iii) MGEs drive genome plasticity, with their distribution structured by bacterial clade and habitat; and (iv) oysters are hotspots of gene flow between vibrios, whose phylopathotypes harbor an unexpected diversity of MGEs, including novel phage-plasmids and phage-satellite-plasmids. Together, these results highlight how host-associated marine environments can couple ecological persistence with genomic diversification.

Our time-series revealed the persistence of major *V. crassostreae* clades and of multiple virulent and temperate phage lineages over several years, with limited genomic divergence despite the open marine setting. Such stability is notable given the strong tidal currents at this intertidal Atlantic oyster farm, which experiences frequent water renewal and would be expected to promote rapid microbial dispersal and turnover. The oysters used in this study were hatchery-reared juveniles that initially harbor very low vibrio loads. Although physically separated from wild oyster beds, they are connected

through tidal water exchange, and their subsequent colonization reflects continuous seeding from the surrounding water column, including inputs from dense wild oyster populations that act as long-term reservoirs of vibrios. At least two non-mutually exclusive scenarios may account for the observed genomic stability of some phage lineages.

First, phage populations may undergo a few replication cycles between sampling points. In this scenario, phages could persist as long-lived viral particles that replicate episodically during seasonal blooms of susceptible hosts. This interpretation is supported by the low observed substitution rates (~$10^{-5}$ substitutions per site per year) and the long-term viability of virulent phages under laboratory storage conditions. These field observations seem consistent with "leapfrog" dynamics, in which lytic phage lineages can persist for extended periods with limited replication and later re-emerge[32]. Persistence of temperate phages can additionally be explained by their vertical transmission as prophages, consistent with the recovery of closely related prophage sequences in samples that are years apart, as well as the predominance of purifying selection across their genomes.

Second, phage replication may be recurrent but strongly shaped by ecological structure. Host availability is constrained by seasonal dynamics, spatial partition between oysters, and the stable organization of *V. crassostreae* into genetically distinct clades. In this context, phage infection is further restricted by receptor specificity and by diverse host defense systems, many of which are encoded on MGEs[12]. Testing these hypotheses will require novel approaches that directly assess where and how phages persist between infection seasons, such as determining their association with wild oyster tissues, sediments, particulate matter, or other environmental reservoirs, and coupling this with direct estimates of replication and decay rates in situ.

The persistence of the same vibrio clades across years enabled us to examine how host and phage lineages−at the vibrio clade-phage genus level relevant for infection−co-occur and fluctuate in abundance within oysters. We confirmed the clear ecological distinction between the environmental clade V1 and the oyster-associated phylopathotypes. Furthermore, multiple phylopathotype clades were frequently detected within the same individual oyster, suggesting that these clades can coexist without strong competitive exclusion or obligate association. One intuitive hypothesis is that phage predation

regulates the abundance and genetic composition of vibrio populations over time and space. Under this scenario, predator-prey dynamics would be expected to generate negative correlations between phage and host abundances, as observed in some systems, such as *V. cholerae* and its virulent phage ICP1[33]. However, while both phage and bacterial abundances fluctuated over the sampling period, we did not observe signatures of phage-driven collapses of bacterial clades or consistent correlations between the frequencies of predator and prey. Several non-exclusive mechanisms may account for this pattern. Some vibrio strains may persist via phage resistance mechanisms or spatial refuges in oyster tissues. Multiple phages can infect the same clade of vibrio, distributing predation pressure, which results in lower correlations between individual phages and bacteria. The coexistence of multiple vibrio strains within oysters, coupled with strain-specific phage susceptibility and rapid evolution of resistance, can generate cryptic population dynamics in which predator-prey interactions are obscured by shifts among genetically distinct subpopulations[7]. Temperate phages switching to lysogeny at high host densities[34] also contribute to reduce or delay, bacterial lysis. Finally, vibrio population dynamics are likely shaped by multiple biotic and abiotic factors, including the dynamics of mobile genetic elements. For example, satellites may moderate the effects of phage predation by reducing viral production rate. Taken together, our results suggest that phages contribute to structuring vibrio populations, but not through simple cycles of dominance and collapse. Instead, phage-host interactions in this system appear consistent with long-term, structured coexistence, in which both antagonists persist despite fluctuations in abundance.

The persistence of multiple bacterial and phage populations in oysters drives their genetic diversification by setting the stage for MGE-mediated genetic exchanges. The oyster environment is characterized by continuous nutrient inflow, extremely high microbial densities, and strong selective pressures imposed by hemocyte immunity, antimicrobial peptides, metals[21,35,36] and abundant phages (this study). Accordingly, we previously showed that genes involved in conjugative transfer and integrases are strongly upregulated during oyster colonization[37], pointing to active MGE circulation in vivo. We found that seawater-derived strains—particularly those from clade V1—have smaller genomes, lack plasmids, and remain consistently susceptible to phages. The relative scarcity of MGEs in these strains may reflect a trade-off favoring reduced metabolic costs in the planktonic portion of the *V. crassostreae* population, as reported for other surface bacterioplankton[38]. In contrast, oyster-associated strains carry larger genomes and significantly more MGEs, consistent with higher rates of genetic exchanges in this habitat. This ecological partitioning mirrors findings in other facultatively host-associated bacteria. For instance, freshwater *Escherichia coli* isolates tend to have smaller genomes and fewer MGEs than host-associated isolates, which carry more plasmids and phages[39]. The small genomes of *Pelagibacter ubique*, an extremely abundant marine species, have been streamlined to become completely devoid of mobile genetic elements[40]. Such parallels suggest that eukaryotic hosts act as hotspots of microbial genome plasticity that facilitate adaptation to both host and phage pressure, whereas aquatic habitats favor streamlined genomes.

The diversity of MGEs in oyster-associated vibrios is likely to further influence adaptive processes because these elements differ in the traits they carry: plasmids tend to contribute to host adaptation and virulence, whereas prophages and satellites are enriched in genes for phage defense or inter-phage competition. MGE diversity may also enable alternative modes of persistence and transmission. Over 6% of the temperate phages are phage-plasmids, combining the capacity for vertical inheritance as plasmids and long-range dispersal within viral particles. Comparable virus-like elements have been described in vibrios, including viruses with double jelly-roll (DJR) capsid[41]. We also identified previously undescribed families of plasmid phage satellites

(Cf-PISP and PISP), expanding the known diversity of MGEs in Bacteria. These elements expand the known diversity of *Caudoviricetes* satellites by revealing a plasmid-based lifestyle rather than chromosomal integration. Unlike classical integrative satellites, these plasmid elements may exist in multiple copies per cell, which would facilitate the hijacking of helper phages for transmission between cells. Similar elements may have been detected through metagenomic sampling in the open ocean[42] as encapsidated particles containing plasmid-like replication genes (e.g., *parA, repA*). The pDolos satellite-plasmid of *Shewanella* was also recently described, although in this case it targets inoviridae and not dsDNA phages[43]. Satellite-plasmids may influence host and phage evolution in distinctive ways, for example, by promoting gene flow between the satellite and plasmid gene pools, or by maintaining higher effective copy numbers than chromosomally integrated elements, which may increase mutation supply[44]. Together, these observations highlight how the ecological context of marine bacteria shapes the diversity, persistence, and evolutionary potential of mobile genetic elements in natural ecosystems.

Our study reveals that even in an open marine system, host-associated environments such as oysters may act as natural laboratories for microbial evolution, where dense interactions and ecological structure favor the circulation and innovation of mobile genetic elements—including phage-plasmids and satellites—while evolutionary change unfolds on ecological timescales.

## Methods

### Sampling of vibrio and phages in a marine environment

Samples were collected from an oyster farm in the Bay of Brest (Pointe du Château, 48° 20′ 06.19″ N, 4° 19′ 06.37″ W) three times per week from June 28 to September 15, 2021. Sampling began when seawater temperatures reached 16 °C, a threshold associated with oyster mortalities[45]. Specific Pathogen-Free (SPF) juvenile oysters[46] were deployed weekly (1000 per batch) from June 1 to August 24, with mortality monitored at each sampling. When batch mortality exceeded 50%, subsequent sampling targeted oysters from the following week's deployment. The study coincided with an ongoing oyster mortality outbreak lasting until September 9.

On each sampling date, 100 live oysters (≤50% mortality) were collected. Hemolymph was extracted, with individual samples (10 oysters) stored at −80 °C for DNA extraction (Hi1–10; 35 dates = 350 DNA samples). The remaining hemolymph (90 oysters) was pooled, centrifuged, and filtered for viral analysis (35 dates = 35 samples). The pellet was resuspended in marine broth (MB), with fractions used for vibrio isolation and hemomicrobiota DNA storage (−80 °C, 35 dates = 35 samples). Seawater (10 L) was size-fractionated[11] with fractions resuspended and processed for DNA extraction (60, 5, 1, and 0.2 μm; 35 dates = 140 samples). Vibrios were isolated from 0.2 μm fractions, and microbiota samples were stored for further analysis (−80 °C, 35 dates = 140 samples). Viral samples were concentrated 1000-fold using iron chloride flocculation[47], suspended in 0.1 M EDTA, 0.2 M MgCl₂, 0.2 M oxalate buffer at pH6, and stored at 4 °C (viruses from seawater, 35 dates = 35 samples). Phages were concentrated using PEG 8000 1X and NaCl 1 M, incubated ON at 4 °C, centrifuged 30 min at 2800 g, and the pellet was resuspended in 500 μl SM buffer (NaCl 100 mM, MgSO₄·7H₂O 8 mM, Tris-Cl 50 mM). Viral DNA was extracted from seawater and plasma samples (two sources, 35 dates = 70 DNA samples).

### Isolation and classification of *Vibrio crassostreae*

Total vibrio isolates from seawater (0.2 μm fraction) and hemolymph were cultured on Thiosulfate-Citrate-Bile Salts-Sucrose (TCBS) agar plates. From each sampling date, approximately 96 colonies were randomly selected from seawater and hemolymph plates and re-isolated once on TCBS. Putative *V. crassostreae* isolates were identified using multiplex PCR with three primer sets (Supplementary Data 18), with colonies used as templates. Isolates were considered positive if at

least two amplicons were obtained. To refine taxonomic identification, isolates were further analyzed by sequencing the *zrgB* gene, which discriminates among *V. crassostreae* phylogroups. Bacteria were grown overnight in marine broth (MB), and DNA was extracted using the Wizard extraction kit (Promega) according to the manufacturer's instructions. The partial *zrgB* gene was amplified using conserved primers (Supplementary Data 18), sequenced via Sanger sequencing (Eurofins), and used to construct a phylogenetic tree.

### Phage isolation

To assess viral predation rates at the strain level, we used our previously established *V. crassostreae* strain collection[12], which includes 153 isolates from the Bay of Brest and Sylt (Germany), spanning clades V1 to V8. These strains served as bait for phage isolation across 35 sampling dates. For each date, we used a mix of 10 µL of seawater viral concentrate (1000×, equivalent to 10 mL of seawater) and 10 µL of plasma derived from a pooled sample of 90 oysters. Phage infections were detected by plaque formation using soft agar overlays on bacterial lawns. In total, 5355 interactions were tested (35 dates × 153 strains).

To refine strain-level predation rate estimates across spatial and temporal dimensions, we focused on host-date combinations that initially exhibited more than 20 PFU per plate, with hosts belonging to a specific clade. These combinations were further tested using either 10 µL of seawater viral concentrate or 10 µL of plasma.

In a previous study[12], we collected up to six plaques per morphotype and found that they were mostly clonal. Therefore, we purified one phage per plaque morphotype and combination (host and date), resulting in a final collection of 1331 phages. Phages were re-isolated through up to three rounds of plaque purification to ensure purity. High-titer stocks (>10⁹ PFU/mL) were prepared via confluent lysis in agar overlays and stored at 4 °C, with an additional aliquot stored at −80 °C in the presence of 25% glycerol. After four years, isolates remained viable, with titers reduced on average by ~1 log (Fig. S33).

We acknowledge that cross-contamination between sampling campaigns could represent a potential problem for our analysis, given the scale of the isolation effort and the long-term storage of phage stocks. Phage isolations and high-titer stock preparations for the 2021 campaign were performed without handling archived 2017 phage stocks, which were stored separately, and a dedicated refrigerator and freezer were reserved for the 2021 collection. The scale of the screening effort (153 host strains across 35 sampling dates) precluded the systematic inclusion of mock controls. However, phages were purified through up to three successive rounds of plaque isolation, reducing the likelihood of carryover contamination. While accidental contamination cannot be formally excluded, the recovery of closely related but non-identical phage genomes across years (with substitution rates higher than those of prophages) argues against simple laboratory carryover.

### Primer design and ddPCR protocol

To identify SNP-dense regions specific to each *V. crassostreae* subpopulation, we used Snippy v4.6.0 (https://github.com/tseemann/snippy) for variant calling, using GV1681 as the reference genome. SNPs were mapped across 604 *V. crassostreae* genomes, including the 447 isolates from this study, with sensitivity and specificity calculated for each subpopulation. Only SNPs in core genes with optimal sensitivity and specificity were retained. SNPs were then treated as nodes in a graph, with edges weighted by proximity, and filtered based on primer design criteria (25-40 bp primer length, 180-300 bp amplicon size, and at least two specific positions per primer). Protein multiple sequence alignments (MAFFT) were converted to CDS alignments, and SNPs were mapped onto these using GV1681 as a reference. Final primer and probe selection were validated through visual inspection. For

phages, primer design was simplified by selecting core genes unique to the subpopulation of interest but absent in other phages.

Droplet digital PCR reactions were performed either in singleplex reactions using an intercalant DNA dye or in multiplex reactions using dye-labeled specific probes. For single target amplification, reactions consisted of 20 µL mixture per well containing 10 µL of ddPCR Evagreen Supermix, 600 nM of primers (Supplementary Data 18), and 5 µL of DNA. For multiple target amplifications, reactions consisted of 20 µL per well containing 5 µL of ddPCR Multiplex Supermix, 900 nM of primers, 250 nM of dye-labeled specific probes (Supplementary Data 18) with either FAM, HEX, Cy5, or Cy5.5, 4 mM of DDT, and 5 µL of DNA. The ddPCR reactions were incorporated into droplets using the QX100 Droplet Generator (Bio-Rad). Nucleic acids were amplified with the following cycling conditions: 5 min at 95 °C, 40 cycles of 30 s at 95 °C and 60 °C for 60 s using Bio-Rad's C1000 Touch Thermal Cycler. For the singleplex reactions, an extra final droplet cure step of 5 min at 4 °C, then 5 min at 90 °C was incorporated. Droplets were read and analyzed using the Bio-Rad QX600 system and QuantaSoft software (version 1.7.4.0917) in "absolute quantification" mode. Only wells containing ≥10,000 droplets were accepted for further analysis. Blank no-template controls were used as absolute negative controls. A threshold of quantification was set at 1 DNA copy per 20 µL reaction, and this threshold was adjusted for each DNA preparation method to account for differences in sample dilution.

Time-lagged autocorrelations were calculated for each vibrio clade and phage genus from the log-ddPCR quantifications averaged across the ten oysters collected per sample date. Unadjusted autocorrelations were calculated with a maximum time lag of ten sampling dates using the acf function of the Python package statsmodels v0.14.5. Unadjusted cross-correlations were similarly calculated between each vibrio clade and phage genus using the ccf function of the same package out to a maximum of ten sampling date lags, both forward and backward in time. Lomb-Scargle periodograms[48,49] were computed from the same DNA quantification data and using ordinal day numbers with the Lomb-Scargle function of Astropy v7.2.0

### *Vibrio crassostreae* genome sequencing and assembly

Of 512 confirmed *V. crassostreae* isolates, 453 (89%) clustered into distinct clades and were selected for genome sequencing, while 59 isolates (11%) were more diverse and excluded from this study. A total of 447 vibrios were successfully sequenced at the Biomics platform of the Pasteur Institute using Illumina® TruSeq™ DNA PCR-Free High Library Prep Kit, IDT for Illumina-TruSeq DNA UD Indexes. Runs were performed by 96 samples with a first validation on MiSeq flow cell micro v2, paired-end 2x150 cycles, next a production run on NovaSeq 1 lane, paired-end 2x150 cycles.

To improve the genome assembly, 20 *V. crassostreae* strains from our previously established *V. crassostreae* strain collection[12], distributed across the phylogeny, were sequenced by PacBio. The PacBio genome assembly was conducted with Unicycler (version 0.4.9)[50], leveraging a hybrid approach that combines both Illumina short reads and PacBio long reads. Post-assembly processing involved Python scripts for replicon identification, where contigs are classified into chromosomes and plasmids, terminal contig sequence extension using raw Illumina reads to ensure completeness, and searching the assembled contigs against the closest relative genomes. The final manual curation step involved identifying and correcting RNA stretches within the assembled contigs, conducting manual alignments to further refine the assembly and ensure its accuracy, resolving any ambiguous bases that may have emerged during the assembly process, and circularizing the contigs to accurately reflect the complete genomic structure.

Vibrio isolates without Pacbio hybrid sequencing data were de novo assembled using Illumina short reads with post-assembly scaffolding as follows. Reads were trimmed using Trimmomatic v0.39

(LEADING:3, TRAILING:3, SLIDINGWINDOW:4:15, MINLEN:36)[51]. De novo read assembly was performed using Spades version 3.15.2 (--careful --cov-cutoff auto -k 21,33,55,77 -m 10). Contig circularization was manually confirmed by inspection of the assembly graph with Bandage v0.9.0[52]. To improve syntenic consistency, the contigs of each strain were scaffolded along the completed assembly of the most closely related of the 20 strains sequenced with PacBio using RagTag v2.1.0 scaffold[53] with minimap2 v2.24 r1122[54].

## Phage genome sequencing and assembly
The methodology for phage DNA extraction, sequencing, and classification followed the protocol detailed previously[12]. Briefly, phage high-titer stocks were concentrated using PEG precipitation, treated with nucleases, and subjected to phenol-chloroform extraction. DNA integrity was assessed via agarose gel electrophoresis and quantified using Qubit. Phage sequencing was performed at the Biomics platform of the Pasteur Institute (Paris, France). DNA was fragmented using Covaris (target size: 500 bp), and libraries were prepared with the TruSeq™ DNA PCR-Free High Library Prep Kit. Due to inefficiency in adaptor ligation, an amplification step of 14 cycles was added using Illumina P7 and P5 primers (IDT). Sequencing was carried out on a MiSeq Micro v2 flow cell with paired-end 2x150 cycles.

Phage isolate reads were trimmed using Trimmomatic v0.39[51] and assembled *de-novo* with SPAdes v3.15.2. Contaminant contigs were filtered using the UniVec Database, and the resulting one-contig phage genome was manually linearized.

## Genome annotation of bacteria, plasmids, phages, and satellites
Genes were predicted and annotated in *V. crassostreae* genomes using bakta v1.9.2[55] with full database v5.1. Except where otherwise indicated, tools that require predicted coding sequences were provided with the corresponding amino acid or nucleotide sequences predicted by bakta.

Annotations in 5 representative high-quality plasmid sequences were manually curated and refined. Conjugative systems and Mob relaxases were predicted and classified using CONJscan v2.0.1 and MOBscan, respectively, with default parameters[56,57].

Sequenced lytic phages, predicted prophages, and phage-satellites were annotated using pharokka v1.7.3[58] using default settings and database v1.4.0. Automatic annotations were subsequently manually refined in certain prophages and phage-satellites. HMMsearch v3.4[59] and the HMM profiles packaged with SatelliteFinder v0.9[29] were used to identify satellite-associated genes. Genes of unknown function were classified by comparing translated amino acid sequences against online databases using the blastp web portal[60] to screen against non-redundant proteins in RefSeq release 229 with default alignment parameters and using the InterProScan web portal to screen against all available protein signature databases[61]. Structural comparisons were conducted by predicting the structure of unknown proteins as monomeric proteins using AlphaFold3 Server to screen against the available structural databases of FoldSeek in 3Di/AA mode[62,63].

## Inference of plasmid presence and transfer in *Vibrio crassostreae*
Twenty plasmids identified in the 20 Pacbio hybrid-assembled genomes were manually classified by inspection of gene content into five plasmid families: pGV, p1, pAlioth, pMintaka, and pMizar. To identify the relationships within each family, core genes were identified as MMseqs2 v15-6f452[64] protein clusters with sensitivity 7.5, minimum coverage 80%, coverage-mode 5, and minimum sequence ID 80%. Core gene nucleotide sequences were aligned with MAFFT v7.526[65] --globalpair, and the unpartitioned concatenated alignment was used to construct a phylogeny for each plasmid using iqtree v2.3.6 with MFP identifying GTR + F + I as the best model by Bayesian information criterion[66,67].

Illumina reads from each sequenced *V. crassostreae* strain were cleaned with fastp v0.24.0 and remapped with minimap2 v2.28 r1209 with flag -x sr to each of the 20 plasmid sequences to assemble hypothetical plasmids of each family. Protein coding regions were predicted from the SAMtools v1.20 consensus[68] remapped sequences with prodigal v2.6.3[69] using the –p meta procedure, as recommended for plasmids. Weighted gene repertoire relatedness (wGRR) was calculated as previously described[70] between each plasmid and the reference used during remapping, and a plasmid was marked as present in a strain if wGRR was greater than 50%. To infer transfer between strains, the plasmid in each group with the highest wGRR was marked as the most closely related plasmid.

Cleaned reads were used to relative extrachromosomal element copy numbers by remapping, as described above, to the corresponding genome assemblies. Read depths were calculated with SAMtools v1.20 coverage. Read depths were normalized to the mean chromosomal read depth in each strain.

## In silico prediction of prophages, phage-satellites
Prophages were predicted with geNomad v1.8.1 end-to-end[22] using database version 1.7 and default settings. The prophages predicted by geNomad as non-chromosomally integrated (direct terminal repeats or no terminal repeats) were manually inspected for circularization in de novo assemblies. As a filter for genome quality, prophages were only retained for analysis if they were predicted *Caudoviricetes* with a genome size greater than 25 kbp, predicted *Tectiliviricetes* with a genome size of greater than 10 kbp, or predicted *Faserviricetes* with a genome size of greater than 4 kbp, in accordance with the literature[23].

Phage satellites were detected using SatelliteFinder v0.9[29] containerized in Apptainer[71] with MacSyFinder v2.0[72] using the predicted protein sequence output of bakta with flags --linear and --ordered_replicon. Predictions were made with each of the four available satellite model definitions bundled with the software: PICI, cf-PICI, PLE, and P4.

## Distribution of genes between mobile genetic elements
Regions of genomic plasticity (RGPs) were identified using the panRGP module[73] of PPanGGOLiN version 2.2.1[74] to identify the persistent and flexible regions of the pangenome in the full corpus of 605 *V. crassostreae* strains. This identified 32,349 RGPs, which were then classified into their MGE types. RGPs were classified as plasmids (3217 RGPs) if they were identified by geNomad as having likely plasmid origin. RGPs were classified as satellites (450 RGPs) if they contained an entire predicted phage satellite segment predicted by SatelliteFinder. RGPs were classified as prophages (4478 RGPs) if they were either identified by geNomad as containing a prophage sequence or by checkV v1.0.3[75] with database release 1.5 as a complete, low, medium, or high-quality prophage containing at least one viral gene, as previously described[12]. Finally, the presence of integrase genes was predicted in the RGPs (1945 RGPs) using bakta, as above, or annotated as unclassified (22,259 RGPs) if none of the above applied.

Anti-phage defense genes were inferred using PADLOC v2.0.0[76] with prodigal v2.6.3 for gene prediction and hmmer v3.3.2 to screen against database release 2.0.0. Biocide and metal resistance genes were identified by screening for the genes included in the experimentally confirmed gene database of BacMet2[77] with MMseqs, with a minimum sequence identity 70% and a minimum coverage 50%. Antimicrobial resistance genes were identified by screening for the genes included in the protein homolog CARD database v4.0.1[78,79] with MMseqs, with a minimum sequence identity 30% and a minimum coverage 50%. Virulence factors were identified by screening for the genes included in the VFDB core dataset[80] with MMseqs with a minimum sequence identity 70% and minimum coverage 50%. These systems were classified by their localization within the flexible genome regions predicted by panRGP. Comparisons of coding density were

performed for the predictions of each dataset using a Kruskal-Wallis H-test with Benjamini-Hochberg correction for false discovery rate and with Dunn's post-hoc test implemented in scipy and scikit_posthocs, respectively.

### *Vibrio crossostreae* core-genome alignment and phylogenetic inference

Bacterial persistent genome phylogeny was conducted using the PanACoTA workflow[81] version 1.3.1-dev2, without applying Mash v2.3 distance filtering[82]. Prodigal version v2.6.3[69] was used for gene prediction. Persistent gene families were thresholded as being present in 90% or greater of all genomes after MMseqs clustering on an 80% protein identity threshold. IQ-TREE v2.0.3[67] was used for phylogenetic tree inference using the core-genome concatenation, applying the GTR model with 1000 bootstrap replicates. The total pangenome comprised 29,988 families, while the persistent genome contained 3,099 families.

The gene presence-absence table and phylogenetic tree from the PanACoTA workflow were used to perform a phylogeny-aware pangenome analysis using panstripe v0.3.1[83] with '"family = gaussian" parameter provided to model fit. Comparisons of pangenome characteristics between subsets of the *V. crassostreae* population were conducted using pruned subtrees of the same base phylogenetic tree to maintain a comparable scale.

Phylogenetic comparative methods, which leveraged tree topology for correction of statistical non-independence, were performed using the phylolm v2.6.5[84] and phytools v2.4-4[85] packages of R v4.4.1. Specifically, continuous data were analyzed using phylANOVA with 10,000 iterations for *p*-value simulation with Holm-Bonferroni post-hoc testing. The association between genome size and the number of MGEs detected was tested with phylolm using a Brownian motion tree model. Presence/absence of MGEs was modeled as a binary trait using phyloglm with a logistic MLE regression.

### Phage and prophage genomic and functional classification

The entire lytic phage collection (including previously-collected isolates) was clustered using the Viral Intergenomic Distance Calculator (VIRIDIC)[16]. VIRIDIC (v1.1) generates a single output file for all blastn results, which can be a limiting factor when dealing with large datasets. To address this, a custom script was developed to split the blastn outputs into smaller, manageable files, allowing for better parallelization and more efficient processing of the 1268 phage genomes. VIRIDIC utilizes fastcluster complete-linkage clustering for fast hierarchical clustering of the resulting distance matrix. The taxonomic assignment of viruses into genus and species ranks was performed based on the intergenomic similarity thresholds defined by the International Committee on Taxonomy of Viruses (ICTV): genus (≥70% similarity), and species (≥95% similarity). Additionally, lysogeny scores were computed for each lytic phage using BACPHLIP v0.9.6[17] using default options. Clustering on intergenomic sequence identity of predicted prophages was performed using VIRIDIC in the same way. Prophages were taxonomically assigned following the ICTV's Viral Metadata Resource release 19 based on marker gene taxonomy during geNomad prediction.

### Inference of substitution rates in persistent lytic and temperate phage species

We considered a lytic phage species to be persistent if it was isolated in both the 2017 and 2021 sampling campaigns. Similarly, we considered a temperate phage species to be persistent if its lysogens were collected in two or more different years. We generated whole-genome alignments between phage genomes of a single phage species using MAFFT G-INS-i. Because the PHI test implemented in PhiPack v1.0[86] inferred evidence for recombination in LS32, Gubbins v3.4[87] was used to mask recombinant regions with default settings for tree construction, model fitting, and sequence reconstruction[88-90]. Final tree inference and inference of substitution rate were performed using BEAST v2.6.3[91] with 100,000,000 iterations and 10% burn-in and a coalescent constant population size model with an exponential relaxed clock rate and a clock rate prior of 1.9 x 10$^{-4}$ site$^{-1}$year$^{-1}$ (the estimated recombination-free substitution rate in a population of lactophages[19]). Path sampling implemented in the model-selection BEAST package[92,93] did not support strict clock rate or Bayesian skyline population size[94,95] as alternative model specifications.

For each persistent lytic or temperate phage species, proteins were clustered on amino acid sequences using MMseqs2 easy-cluster with sensitivity of 7.5, coverage mode 5, and 80% minimum coverage. Single-copy orthologs were aligned on their amino acid sequences with MAFFT L-INS-i and back-translated to codon alignments using pal2nal.pl v14.1[96]. The CODEML package of PAML v4.10.9[97] was run using the codon alignment and the species phylogenies inferred above to infer support for positive selection for each gene and to determine specific codons under selection in each codon alignment by comparing the fits of the M2a and M1a site model scenarios. Genes with a global signature of selection were identified using a likelihood ratio test to compare these two nested models using a Chi-squared test with two degrees of freedom and alpha of 0.01. For these genes, the percentage of codons under selection was calculated by identifying the number of codons where the Bayes Empirical Bayes score was above 0.8.

### Satellite-plasmid detection and comparison

SatelliteFinder was used to characterize TG54, a 15 kbp contig identified by geNomad as a putative *Caudoviricetes*. Though SatelliteFinder did not identify TG93 as a satellite, inspection of its annotated gene content led us to identify it as a cfPICI-like element. TG54 (cf-PISP) encodes a PFam family PF05065 capsid protein, and TG93 (PISP) encodes a PFam family PF03864 CapsidPE protein. These HMM profiles were downloaded from the InterPro repository[98,99]. HMMsearch v3.4 [http://hmmer.org/] was used with default parameters to identify homologous proteins of each capsid family in these new elements, in the phage-satellite sequences included with the supplementary data of ref. 100 and in the lytic vibriophages cataloged in this manuscript. For each capsid family, gene sequences were aligned with MAFFT L-INS-i, and a maximum-likelihood tree was inferred using IQtree with the Q.pfam+F + I + R model specified. The most similar phage-satellite to each of our elements was chosen using these capsid phylogenies and used, together with the prototypical PICI or cfPICI with the most experimental characterization, for comparisons of genetic organization. Comparisons were produced by computing bi-directional best hits with MMseqs and visualized with pygenomeviz v1.4.1 (https://github.com/moshi4/pyGenomeViz).

We looked for similar elements in public repositories by screening for similarities to the capsid or capsidPE and the ParA protein sequences to identify other species carrying plasmid-satellites. We searched for similar sequences using the blastp web portal in non-redundant proteins in RefSeq release 229 with default alignment parameters. From the top 100 hits for each search, the genetic contexts of the target protein were manually inspected for similarity to the plasmid-satellite gene catalog and for the size of the contig and its integration into the host chromosome. Each homolog was downloaded and re-annotated with pharokka v1.7.3 using phanotate for gene prediction, with the expectation that coding density and gene length distribution in these elements are best modeled as phage-like. Syntenies were produced as above.

### Reporting summary

Further information on research design is available in the Nature Portfolio Reporting Summary linked to this article.

## Data availability

The genome sequences of phages and *V. crassostreae* isolates generated in this study have been deposited in the European Nucleotide Archive (ENA) under BioProjects PRJEB81325 (phages) and PRJEB67885 (bacterial genomes). Accession numbers are listed in Supplementary Data 1 and 4. Column E of Supplementary Data 1 ("Accession") refers to the submitted GenBank assembly of each newly isolated phage and refers to newly published data. Column G of Supplementary Data 1 ("Host Accession") refers to the host strain of *V. crassostreae* used for the isolation of the phage and as consequence of the experimental design, refers to previously published data[12]. Column D of Supplementary Data 4 ("Accession") refers to the submitted Genbank assembly of each newly isolated *V. crassostreae* strain and refers to newly published data.

The Roscoff Culture Collection (https://roscoff-culture-collection.org/), permits access to the biological material (phages and bacteria isolated in the present study, except when stated otherwise, and can be provided from Le Roux lab at Montreal) upon request, with accession numbers listed in Supplementary Data 1 and 4. Source data are provided with this paper.

## Code availability

Custom code used in this study is publicly available on Zenodo (https://doi.org/10.5281/zenodo.18717761)

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

## Acknowledgements

We thank Bruno Petton (Ifremer Brest) for providing oyster juveniles and overseeing in situ experiments, which made the time-series sampling possible. We are grateful to Chloé Berger, Pauline Daszkowski, Justine Groseille, Théo Foutel Rodier, Étienne Levêque, and Mariam Mamba (GV team, Station Biologique de Roscoff, France) for their assistance during sampling and collections, and Carine Diarra (Le Roux lab, Montréal) for technical help. We thank Jorge Moura de Sousa for helpful discussions concerning the biology and classification of phage-satellites and Julien Guglielmini for help running wGRR/GRIS. We thank Léa Verena Zinsli and Otto X. Cordero for valuable comments on the manuscript, and Illumina for reduced pricing of sequencing kits. Editorial support from Stephen Matheson (Life Science Editors) is also warmly acknowledged. This research was enabled in part by the computational resources of Calcul Quebec and the Digital Research Alliance of Canada.

This work was supported by funding from the European Research Council (ERC) under the European Union's Horizon 2020 research and innovation program (grant agreement No. 884988, Advanced ERC "DYNAMIC"), the Canada Excellence Research Chairs Program (CERC-2022-00051), and the Fonds de recherche du Québec – Nature et technologies (FRQ-NT, Fonds des leaders John-R.-Evans, grant 44584) awarded to FLR. Substantial support was provided by the Agence Nationale de la Recherche (ANR-20-CE35-0014 "RESISTE") and the Fonds UdeM-Pasteur pour la découverte de nouveaux antibiotiques et antibactériens, a philanthropic fund at the Courtois Institute in Biomedical Innovation, Faculty of Medicine, University of Montreal, to EPCR and FLR. Biomics Platform, C2RT, Institut Pasteur, Paris, France, is supported by France Génomique (ANR-10-INBS-09) and IBISA (EPCR, LM, and MM).

## Author contributions
F.L.R. conceived the study, F.L.R. and E.P.C.R. supervised the project, and secured funding. K.C., D.P., D.C.G., Y.L., L.M. and F.L.R. conducted the experiments. D.P. and D.C.G. contributed equally. J.L., D.G., C.B. and E.P.C.R. performed the genomic analyses. J.L., KC, DP, DCG, MM, EPCR, and FLR analyzed the data. J.L., E.P.C.R. and F.L.R. wrote the manuscript.

## Competing interests
The authors declare no competing interests.
