## [Transparent Peer Review file · Nature Communications]

Complex temporal dynamics of phage-bacteria populations in an animal-associated marine system

Corresponding Author: Professor Frédérique Le Roux

Version 0:

Reviewer comments:

Reviewer #1

(Remarks to the Author)

Overall: This is an exciting dataset and paper examining the complex interplay between viruses, vibrio hosts, MGEs (in part) and the oyster microenvironment. The data suggests persistent relationships between viruses and their hosts, perhaps 'filtered' by requirements to thrive in the oyster environment. The paper also needs refinement, especially when it comes to the time-series analysis and part of the modularity/network analysis. Nonetheless, with changes, I think this paper has the chance to broaden interest in factors that drive dynamic stability in a coevolutionary, environmental context. I enjoyed the paper.

I organize comments below based on concepts in the paper and then finish with a series of specific comments vis a vis the writing in the Abstract/Introduction.

Stability

The authors claim remarkable stability. But, Figure 1 shows dynamic variability. I am trying to reconcile these ideas. Because some of the viruses persist over multiple years (as shown in Figure 2), but some do not. I agree that finding the same virus multiple years later is exciting! But the line 'virulent phages persist over multiple years' would be strengthened by providing some sense of capture-recapture information or some measure of the balance between dynamic change and clonal genome preservation. More clarity is welcome here, the findings are very interesting. I also find L156 confusing and not sure what this hypothesis really means. If bacteria coevolve wouldn't that disfavor rather than favor persistence of phage lineages? Moreover, when I look at the phylogenies in the SI, it seems that 2017 and 2021 lineages often group separately which does not seem consistent with stability but instead, with change. Line 477-478 again suggest a central conclusion that applies to ALL the phage, but I think the authors find that it applies to SOME of the phage, it's already an exciting result, it just needs to be written with greater precision.

Modularity

L33, the authors write "The phage-vibrio infection network remained modular..." The use of a term infection network implies cross-infection measurements between diverse phage and vibrio as part of a bipartite network. Whereas Figure 2 shows a 'modular' isolation network that is NOT an infection network. There is no data here that show who infects whom. The authors have used host strains for isolation, but I do not think that they have conducted a full cross-strain analysis, have they? If so, where is that data? I think this is an intergenomic stability network where the host clade is included for convenience but not part of the modularity/clustering algorithm. I find Figure S1 quite helpful, though the caption should be extended with more details and the colorbar should clearly denote what the matrix shows. Figure 2 involves cutoffs that imply 0 "connectivity" between phage. I just want to clarify here – are you implying that there is 0 'intergenomic similarity' between these viral groups. Is that true? I am confused – perhaps by VIRIDIC, the present use of VIRIDIC, or the visualization.

Gaps in quantitative analysis

I don't understand the basis for claims in lines 176-179: "We next quantified the abundance of phage genera infecting these clades in the same samples (Figure S11). Phages infecting V2 (LG12), V5(LG48), and V8 (LG3) showed periodic blooms in plasma, closely tracking host abundance. In contrast, phages infecting the environmental clade V1 (LG24, LG26, LG31) displayed more stable levels across habitats." The reality is that all of these time series are fluctuating and the authors are asserting features "periodic blooms" and "more stable levels" without conducting any kind of non-parameter test for periodicity or accounting for the fact that time series can exhibit fluctuations that appear spuriously periodic (this is true even

for random walk time series when not accounting carefully for the build-up of persistent deviations). I don't see what the authors see in Figure S11 and the authors have not provided any quantitative analyses to support their claims. This needs to be strengthened and examined carefully. Likewise, claims on 184-5 about coincident phage-host peaks are not demonstrated and yet the authors infer from this data the "dependence of phages on host presence". This is an unsupported claim, especially in light of the next sentence and analysis.

Predator-prey dynamics

Figure 3 is under-utilized. The Figure title says 'Predator-prey dynamics' but the authors do not conduct a rigorous, time-shifted analysis of the pairwise time series to see if, in fact, the predator and prey time series have characteristic time-shifted (usually quarter phase) oscillations. Do they? Or do they not? Even if a lytic phage can kill a host, that does not mean that the time series exhibits predator-prey like dynamics. The authors are moving too quickly through their time-series data, making leaps to conclusions that require further analysis – there is a real opportunity to do something meaningful here. Here are a few older papers that treat this issue

<https://pubmed.ncbi.nlm.nih.gov/17803356/>

<https://pubmed.ncbi.nlm.nih.gov/24799689/>

The lines between 422-425 about 'negative correlations' is incorrect. The authors should look above to frame their expectations vis a vis time series of predator and prey. Nor should the authors expect the virulent phage drive their populations towards sharp decline, viruses are intracellular obligate parasites and the expectation should be that they lead to oscillations in population abundances; though many factors can modify that expectation (see above).

Virulent vs. Temperate

Lines 282-286 tell a very interesting story, which I were learning more about it. Likewise the MGE data is is very interesting though I am less clear on how it impacts the central phage-vibrio story (obviously the phage-plasmid part is relevant, perhaps it's simply a matter of the complexity of the data here).

Writing

The authors should reduce their use of adverbs and clean up the writing, especially in the Abstract/Introduction.

L28: Phages are typically viewed as very rapidly evolving biological entities. Why 'very' and this is an odd way to start.

Typically viewed by whom?

L29: 'establish long-term genetic stability'. I don't know what this means. With whom? As package genomes? As genomes inside hosts? As populations? This setup of the abstract is unclear and confusing.

L30: The authors write 'We addressed this eco-evolutionary question..' But the authors have not specified a question in the first two lines. The writing is awkward here, which is not ideal given these are the first 3 lines of an excellent paper. The abstract must be rewritten.

L32: Surprisingly – to whom? 'very persistent' vs. 'persistent'. The authors have not established a case of instability for the reader to be surprised. Reduce this kind of editorializing and state the findings please.

Overall, the abstract seems like the paper will have interesting data (it does!) but does not explain in plain terms what the question is, what the findings are, and what readers will learn.

L46: very diverse -> diverse

L47: have rapid turnovers -> turnover rapidly

First paragraph of Introduction is also confusing, are viruses diverse and changing or not so diverse and stable? I think the authors know what they want to say but are either rushing the text or missing an opportunity to clarify the tension at play here. Needs clarification.

L58: The authors use the word 'questions' but do not have questions in the first paragraph. L72-75 finally have questions – and they are clear! This is where the paper really takes off.

Reviewer #2

(Remarks to the Author)

I found this study interesting for its tracking of bacterial and phage genomic information across multiple levels of ecological organization. It presents the and supports well the reasonable hypothesis that host animals such as oysters serve as islands/reservoirs of microbial and genetic persistence in open-water systems. Additionally it is intriguing for its proposal of a new kind of mobile genetic element, cf-PISPs. The data are rich and the reasoning largely lucid, detailed, and sound. I would recommend this manuscript for publication.

Line 205: the idea that the environment inside oysters promotes Vibrio-phage coexistence (separate from its promotion of genomic plasticity and genetic exchange) is exciting, however the reasoning would be stronger if the authors could show clade-level bacteria-phage abundance was more correlated in oysters as compared to in seawater, i.e. Fig. 3B but for seawater as well. Alternatively, perhaps I mistook the authors' intent, in which case that might be made more clear.

Fig. 3B: it would be good to see the CI/SD/distribution of data as ribbons or intervals (cf. Fig. S12A), as it is difficult to tell anything about the shape of the distribution from the dense individual dots.

Fig. 5A: 'no terminal repeats' are represented as diamonds in the figure, but the caption says triangles.

Fig. S1: annotate XY axes to make clear which one is hosts and which phages.

- William K. Chang

(Remarks to the Author)

Overall assessment

This is a rich, carefully executed study that combines long-term field sampling, dense culture-based phage isolation, genomics, and MGE analyses to address how phage–host lineages behave over years in an open marine system. The dataset is genuinely impressive and many of the conclusions about lineage persistence, modular infection structure, and the diversity of MGEs in oyster-associated *Vibrio crassostreae* are well supported. I am enthusiastic about publication after revision.

My main reservations concern (i) the repeated framing of “ecological constraints” and “constraints on antagonistic coevolution” as the primary explanation for phage genomic stability, and (ii) the assumption that lytic phages are engaged in ongoing arms races rather than spending much of their time in a low-replication, persistence mode. These issues are conceptual rather than technical, and can largely be addressed by tempering language and more fully considering alternative explanations that parsimonious and align with the data presented than the “constraints” explanation does.

Comments

1. “Ecological constraints” vs “limited replication / passive persistence”

The title, abstract, and Discussion repeatedly assert that “nested ecological constraints” or “environmental constraints” stabilize viral lineages and constrain evolution (e.g., Abstract lines 38–42; Discussion lines 400–417).

However, the primary observations are:

- o For several virulent species, there are phage genomes that are identical across four years (e.g., LS29), and others that differ by only a handful of SNPs.
- o For many temperate species, there is even less change, including prophages identical across two decades. These patterns are very naturally explained by low cumulative replication and/or long-lived particles, rather than by strong constraints acting on rapidly evolving, frequently replicating populations. If the lytic phages were engaged in continuous arms races but constrained, one would expect:
 - o measurable substitution rates with at least synonymous changes accumulating genome-wide, and
 - o non-synonymous changes concentrated in interaction loci.

Instead, for some lineages there are no mutations at all, which seems more consistent with very few replication rounds between sampling points than with strong purifying selection under active coevolution.

Suggestions:

o Please soften or rephrase the claim that ecological constraints per se are responsible for genomic stability. For example, the title could shift from “Ecological constraints foster...” to “Ecological context and low replication facilitate...” or similar.

o In the Discussion, explicitly contrast two scenarios:

1. Active phage–host arms races whose outcomes are tightly constrained; vs.
2. Phage populations that spend most of the time in a low-replication persistence state (e.g., in sediments, on particles, or in low-MOI infections), with occasional bursts during host blooms.
 - o Make clear that your data cannot distinguish between these mechanisms at present, and that passive persistence is a parsimonious explanation for the complete absence of sequence change in some lineages.

2. Assumption of arms-race coevolution for virulent phages

Throughout, lytic phages are described as being in antagonistic coevolution or “arms races” with their hosts. Yet several lines of evidence argue against strong ongoing arms races in this specific system:

- o Abundances of phages and hosts are often positively correlated within oysters and across time; there is no clear predator–prey lag or signature of recurrent selective sweeps.
- o For some virulent species, you find no substitutions across years, rather than the accumulation of synonymous (and occasionally non-synonymous) changes expected under constrained arms-race dynamics.
- o You show experimentally that virulent phages retain viability after years of storage with only ~1 log drop in titer, suggesting that long-lived particles might also be possible in the environment.

Taken together, these observations look more compatible with phage persistence with limited replication than with continuous arms races.

3. dn/ds (or related) analyses could help distinguish hypotheses

In Figure 5B you compare substitution rates of virulent vs temperate phages, but the manuscript does not assess the distribution of non-synonymous vs synonymous substitutions.

- o If ecological or functional constraints are indeed limiting evolutionary outcomes despite ongoing mutation, one would expect $dn/ds < 1$ genome-wide, with especially low ratios in critical structural and receptor-binding genes.
- o Conversely, if there is very little replication, dn and ds are both near zero and the ratio is difficult to interpret.

Suggestions:

- o For lineages with sufficient polymorphism (e.g., LS32, perhaps some prophage species), please compute dn/ds or an analogous metric (e.g., pN/pS) at least at the whole-genome level, and ideally partitioned into structural vs non-structural genes.
- o Explicitly discuss how these results bear on the “constraints vs persistence” question.
- o In spots where you currently invoke constraint based purely on low substitution rates (e.g., Discussion lines 411–417), consider replacing those claims with interpretations supported by dn/ds , or softening if the data are inconclusive.

4. Interpretation of “unexpected ecological forces”

Related to comments 1–3, some parts of the Discussion imply that there must be highly unexpected ecological forces maintaining stability (e.g., nested ecological filters constraining diversification).

In fact, the idea that phage particles can persist for long times with minimal replication—and that phage–host contact may be intermittent—is not particularly exotic and is consistent with your own laboratory stability data. The danger is that readers may leave thinking they should look for special “constraint mechanisms” rather than for mechanisms that permit viable phage persistence for years (e.g., low decay in cold sediments, adsorption to particulate matter or biofilms, repeated “reseeding” from refugia).

Suggestion:

Reframe these sections to emphasize that:

- o Your data highlight the surprising longevity of some virulent phage lineages;
 - o The main mechanistic question for future work is how phages remain viable for years with low apparent replication, not necessarily what exotic ecological constraints suppress evolution during an arms race.
- This is an important distinction because it will shape how follow-up studies are designed.

5. Contamination controls for identical genomes across years

The conclusion that identical phages were recovered four years apart is central, but the methods do not explicitly describe contamination controls between the 2017 isolates and the 2021 sampling campaign. The Materials and Methods section explains how plaques were purified and how high-titer stocks were stored, and notes that 2017 stocks remained viable after four years.

Given that:

- o 2017 phage stocks were stored in the same lab and remained viable; and
 - o 2021 phage isolations, amplifications, and sequencing appear to have been done in the same general setting,
- there is a non-trivial risk that a stored stock could have been inadvertently introduced into 2021 plates (e.g., via aerosols, shared pipettes, or reuse of phage stocks as positive controls).

Suggestions:

- o Explicitly state how cross-contamination between 2017 and 2021 isolates was prevented (physical separation of stocks and workspaces, separate freezers, no use of archived phages as positive controls, etc.).
- o If available, mention any negative controls that were processed in parallel (e.g., mock overlays with buffer) and their results.
- o If contamination cannot be definitively ruled out, acknowledge this limitation when claiming “identical genomes 4 years apart,” even if you regard it as unlikely.

6. Claims about unprecedented scale

The manuscript describes the dataset as “unprecedented” in scale (e.g., Abstract lines 31–32, Results lines 76–82). While the combination of >1,000 phages and 600 host genomes for a single species is indeed impressive, prior work (notably from Polz and collaborators) has produced very fine-scale, high-frequency time series of marine phage–host dynamics, in some cases with denser temporal sampling.

Suggestion:

Soften the language to something like “a large, well-resolved dataset” or “a uniquely dense dataset for a single bacterial species–phage assemblage,” and explicitly acknowledge prior high-resolution time-series work (e.g., Polz lab) as complementary.

7. Connection to experimental work showing long-lived lytic lineages (Leap-Frog dynamics)

You note that experimental coevolution can produce rapid diversification (e.g., for *Synechococcus* cyanophages), but there is also experimental literature where lytic phage lineages persist cryptically for long periods despite the potential for rapid evolution (e.g., Leap-Frog dynamics experiments, where “hidden” lineages re-emerge after hundreds of generations).

Suggestion:

- o In the Introduction or Discussion (perhaps around lines 50–57 or 380–387), cite the Leap-Frog Dynamics work and point out that your field observations of long-lived virulent lineages are conceptually similar, and could potentially be reproduced in controlled lab systems.

8. Figure 3B readability

The x-axis tick labels in Figure 3B are very hard to read in the current form. Please increase font size or reduce the number of ticks.

9. Host range bias from using archival strains as bait

You use a panel of 153 archival *V. crassostreae* strains as bait for phage isolation in 2021. This is sensible for consistency, but it implicitly biases detection toward phages that infect those specific strains.

Please note this as a limitation: phages specialized on 2021-specific lineages that are poorly represented in the archival collection might be under-sampled.

Recommendation

In summary, I support publication of this manuscript after revision. The work presents impressive and novel data, and the conceptual advances around phage and MGE diversity in an oyster-associated system are substantial. Addressing the points above will make the paper both more rigorous and more useful to researchers who build on this system.

Version 1:

Reviewer comments:

Reviewer #1

(Remarks to the Author)

This was a strong paper before and stronger now with appropriate controls and caveats. The authors have done an exceptional job in taking reviewer comments and addressing issues, correcting claims when necessary, and writing a paper that is easier to interpret and will be more durable. I commend them. I could quibble here and there, but won't. Instead, I make one suggestion.

On page 13, the authors interpret their findings by writing: "Two non-mutually exclusive scenarios may account for the observed genomic stability of some phage lineages." I am reminded of a column "In theory" by the Nobelist Sydney Brenner in Current Biology (can be found here <https://tavarnarakislab.gr/news/Loose-Ends-and-False-Starts.pdf>), in which Brenner writes:

I recall a meeting in the 1970s where a speaker presented two different models of transposition, which we can call A and B. The climax of the talk came when the speaker triumphantly declared that there were only two possibilities: "Either A is right and B is wrong, or B is right and A is wrong." He had to be reminded that he had overlooked a third possibility which was that they were both wrong.

I would simply caution that given the complexity of the data, the two scenarios the authors envision are not the only ones and it would be important to remind the reader that there may be new eco-evolutionary mechanisms at play. The finding of stability is precisely one of the reasons I think this paper will attract readers and new hypotheses.

Reviewer #2

(Remarks to the Author)

Authors have addressed my comments on the original manuscript.

Reviewer #3

(Remarks to the Author)

I'm happy with the revisions and responses to my review. Because of time restrictions, I was unable to evaluate responses to the other reviewers.

REVIEWER COMMENTS

We sincerely thank all three reviewers for their careful evaluation of our manuscript and for their constructive and thoughtful feedback. Their comments significantly improved the clarity of the analyses, strengthened the interpretation of the data, and helped refine the conceptual framing of phage-host persistence, dynamics, and mobile genetic element diversity in this system. We greatly appreciate the time and expertise invested in this review process.

Reviewer #1 (Remarks to the Author):

Overall: This is an exciting dataset and paper examining the complex interplay between viruses, vibrio hosts, MGEs (in part) and the oyster microenvironment. The data suggests persistent relationships between viruses and their hosts, perhaps 'filtered' by requirements to thrive in the oyster environment. The paper also needs refinement, especially when it comes to the time-series analysis and part of the modularity/network analysis. Nonetheless, with changes, I think this paper has the chance to broaden interest in factors that drive dynamic stability in a coevolutionary, environmental context. I enjoyed the paper.

I organize comments below based on concepts in the paper and then finish with a series of specific comments vis a vis the writing in the Abstract/Introduction.

We thank Reviewer #1 for the detailed and rigorous evaluation of the manuscript. These comments were instrumental in improving the clarity of the time-series analyses, refining the interpretation of stability versus variability, and strengthening the quantitative support for our conclusions. In particular these suggestions led us to implement additional autocorrelation and spectral analyses and to substantially revise the framing of predator-prey dynamics. We greatly appreciate the careful reading and constructive guidance.

Stability

1.1 The authors claim remarkable stability. But, Figure 1 shows dynamic variability. I am trying to reconcile these ideas. Because some of the viruses persist over multiple years (as shown in Figure 2), but some do not. I agree that finding the same virus multiple years later is exciting! But the line 'virulent phages persist over multiple years' would be strengthened by providing some sense of capture-recapture information or some measure of the balance between dynamic change and clonal genome preservation. More clarity is welcome here, the findings are very interesting.

We thank the reviewer for highlighting the need to better articulate the balance between temporal dynamics and lineage persistence. We agree that our data reveal both substantial ecological variability and long-term genomic stability, and that this duality was not sufficiently explicit in the original version.

Figure 1 captures temporal fluctuations in phage recovery and host-specific predation intensity, but does not address phage genotypic identity. In contrast, Figure 2 and the associated phylogenetic analyses focus on genomic relatedness among phages isolated across dates and years. Together, these analyses show that while phage presence and abundance vary a lot, some viral lineages persist over multiple years with highly conserved genomic architectures.

Importantly, this persistence does not imply the absence of evolutionary change. Even within persistent species, we observe local sequence diversification, including SNP accumulation and recombination hotspots, indicating ongoing genetic change superimposed on global lineage stability. We have revised the text to explicitly acknowledge this coexistence of stability and change, and to clarify that long-term persistence applies to a subset of phage lineages rather than uniformly across the community.

1.2 I also find L156 confusing and not sure what this hypothesis really means. If bacteria coevolve wouldn't that disfavor rather than favor persistence of phage lineages?

We agree that the original phrasing was misleading. We have removed the reference to coevolution and now introduce this section as an exploration of phage–host abundance dynamics enabled by the clade-specific association between phage genera and their hosts.

1.3 Moreover, when I look at the phylogenies in the SI, it seems that 2017 and 2021 lineages often group separately which does not seem consistent with stability but instead, with change. Line 477-478 again suggest a central conclusion that applies to ALL the phage, but I think the authors find that it applies to SOME of the phage, it's already an exciting result, it just needs to be written with greater precision.

We now explicitly state that long-term genomic conservation applies to a subset of phage species, and that this occurs alongside substantial temporal and genomic variability. Note that we lack outgroups to assess if the most recent lineages have higher substitution rates from the ancestor, as expected. Still, although in many cases the two sampling periods are clustering apart the substitution rates separating them are very small, consistent with the existence of a diverse population that persisted in the location.

1.4 L33, the authors write “The phage-vibrio infection network remained modular...” The use of a term infection network implies cross-infection measurements between diverse phage and vibrio as part of a bipartite network. Whereas Figure 2 shows a ‘modular’ isolation network that is NOT an infection network. There is no data here that show who infects whom. The authors have used host strains for isolation, but I do not think that they have conducted a full cross-strain analysis, have they? If so, where is that data? I think this is a intergenomic stability network where the host clade is included for convenience but not part of the modularity/clustering algorithm.

We thank the reviewer for this important clarification. We agree that referring to an “infection network” was misleading in the context of this study, as we did not reconstruct a cross-infection matrix here.

We have therefore revised both the text and Figure 2A to avoid this terminology. We now describe the data as phage isolation patterns based on intergenomic clustering (VIRIDIC), with phage genera annotated by the *V. crassostreae* clade used for isolation. Importantly, host clade information is not used in the clustering itself, but only for visualization.

We now explicitly state that phage were predominantly associated with a single *V. crassostreae* clade, the one used for its isolation (Figure 2A). This pattern is consistent with the clade-specific infection modules previously identified by cross-infection assays (Piel et al. 2022), although no cross-infection matrix was performed in the present study.

We believe this revised phrasing more accurately reflects the experimental design and avoids over-interpreting the isolation data as an infection network.

1.5 I find Figure S1 quite helpful, though the caption should be extended with more details and the colorbar should clearly denote what the matrix shows.

The concerns raised for the former Figure S1 are now addressed in the revised Figure 2A, which includes an expanded caption and explicit annotation of the VIRIDIC matrix.

1.6 Figure 2 involves cutoffs that imply 0 “connectivity” between phage. I just want to clarify here – are you implying that there is 0 ‘intergenomic similarity’ between these viral groups. Is that true? I am confused – perhaps by VIRIDIC, the present use of VIRIDIC, or the visualization.

This concern is addressed by the revised Figure 2A, which now directly displays VIRIDIC intergenomic similarity and no longer implies zero connectivity between viral groups.

1.7 Gaps in quantitative analysis

I don't understand the basis for claims in lines 176-179: “We next quantified the abundance of phage genera infecting these clades in the same samples (Figure S11). Phages infecting V2 (LG12), V5(LG48), and V8 (LG3) showed periodic blooms in plasma, closely tracking host

abundance. In contrast, phages infecting the environmental clade V1 (LG24, LG26, LG31) displayed more stable levels across habitats.” The reality is that all of these time series are fluctuating and the authors are asserting features “periodic blooms” and “more stable levels” without conducting any kind of non-parameter test for periodicity or accounting for the fact that time series can exhibit fluctuations that appear spuriously periodic (this is true even for random walk time series when not accounting carefully for the build-up of persistent deviations). I don’t see what the authors see in Figure S11 and the authors have not provided any quantitative analyses to support their claims. This needs to be strengthened and examined carefully.

We improved the study by autocorrelation analyses of both bacterial and phage time series and Lomb–Scargle periodogram analyses (Figure S12-14). In light of the results, we have re-interpreted our data to reflect the lack of periodicity.

1.8 Likewise, claims on 184-5 about coincident phage-host peaks are not demonstrated and yet the authors infer from this data the “dependence of phages on host presence”. This is an unsupported claim, especially in light of the next sentence and analysis.

We removed this sentence according to the reviewer’s instruction.

Predator-prey dynamics

1.9 Figure 3 is under-utilized. The Figure title says ‘Predator-prey dynamics’ but the authors do not conduct a rigorous, time-shifted analysis of the pairwise time series to see if, in fact, the predator and prey time series have characteristic time-shifted (usually quarter phase) oscillations. Do they? Or do they not? Even if a lytic phage can kill a host, that does not mean that the time series exhibits predator-prey like dynamics. The authors are moving too quickly through their time-series data, making leaps to conclusions that require further analysis – there is a real opportunity to do something meaningful here. Here are a few older papers that treat this issue

<https://pubmed.ncbi.nlm.nih.gov/17803356/>

<https://pubmed.ncbi.nlm.nih.gov/24799689/>

The lines between 422-425 about ‘negative correlations’ is incorrect. The authors should look above to frame their expectations vis a vis time series of predator and prey. Nor should the authors expect the virulent phage drive their populations towards sharp decline, viruses are intracellular obligate parasites and the expectation should be that they lead to oscillations in population abundances; though many factors can modify that expectation (see above).

We have formalized our time-series analysis of the ddPCR-based population quantifications (Figures S13 and S14) and have re-considered our results in these respects. We did not detect periodicity in our Lomb-Scargle models (Figure S14) and thus are unable to directly apply the models developed in the suggested manuscripts but have included them as valuable discussion points.

1.10 Virulent vs. Temperate

Lines 282-286 tell a very interesting story, which I were learning more about it. Likewise the MGE data is is very interesting though I am less clear on how it impacts the central phage-vibrio story (obviously the phage-plasmid part is relevant, perhaps it’s simply a matter of the complexity of the data here).

We thank the reviewer for this comment and agree that the diversity of MGEs adds complexity to the manuscript. We would like to clarify that the analysis of MGEs is central to our phage-Vibrio story, rather than peripheral.

In this system, MGEs are strongly enriched in oyster-associated *Vibrio* populations and play a key role in shaping phage-host interactions. Temperate phages, phage-plasmids, and phage satellites directly affect both bacterial susceptibility and phage propagation, thereby influencing coexistence, persistence, and apparent stability of phage lineages.

While phage-plasmids are one clear example of this link, the newly described satellite-plasmids represent an equally important and previously unrecognized component of the phage-MGE continuum. To our knowledge, these are the first examples of satellites

maintained as plasmids for dsDNA phages, and their discovery directly expands current models of phage-host and phage-phage interactions.

We believe that the clarifications made throughout the manuscript—particularly regarding isolation patterns, temporal dynamics, and the distinction between lineage persistence and change—now better integrate the MGE analyses into the central ecological and evolutionary framework of the study.

Writing

The authors should reduce their use of adverbs and clean up the writing, especially in the Abstract/Introduction.

We hope that the revisions to our writing address the reviewer's concerns.

1.11 L28: Phages are typically viewed as very rapidly evolving biological entities. Why 'very' and this is an odd way to start. Typically viewed by whom?

We agree and have removed the word "very" to avoid unnecessary emphasis and improve clarity.

1.12 L29: 'establish long-term genetic stability'. I don't know what this means. With whom? As package genomes? As genomes inside hosts? As populations? This setup of the abstract is unclear and confusing.

1.13 L30: The authors write 'We addressed this eco-evolutionary question..' But the authors have not specified a question in the first two lines. The writing is awkward here, which is not ideal given these are the first 3 lines of an excellent paper. The abstract must be rewritten.

1.13 L32: Surprisingly – to whom? 'very persistent' vs. 'persistent'. The authors have not established a case of instability for the reader to be surprised. Reduce this kind of editorializing and state the findings please.

1.14 Overall, the abstract seems like the paper will have interesting data (it does!) but does not explain in plain terms what the question is, what the findings are, and what readers will learn.

The whole abstract has been rewritten

1.15 L46: very diverse -> diverse

Edited

1.16 L47: have rapid turnovers -> turnover rapidly

Edited

1.17 First paragraph of Introduction is also confusing, are viruses diverse and changing or not so diverse and stable? I think the authors know what they want to say but are either rushing the text or missing an opportunity to clarify the tension at play here. Needs clarification.

We agree and have revised the first paragraph of the Introduction to more explicitly articulate the tension between rapid viral diversification and long-term genomic stability, thereby clarifying the conceptual framework motivating this study.

1.18 L58: The authors use the word 'questions' but do not have questions in the first paragraph. L72-75 finally have questions – and they are clear! This is where the paper really takes off.

We agree and have revised the paragraph to ensure that the term "questions" is used only where the specific research questions are explicitly stated, thereby improving clarity and flow.

1.19 Need to rewrite the abstract

The whole abstract has been rewritten

Reviewer #2 (Remarks to the Author):

I found this study interesting for its tracking of bacterial and phage genomic information across multiple levels of ecological organization. It presents the and supports well the reasonable hypothesis that host animals such as oysters serve as islands/reservoirs of microbial and genetic persistence in open-water systems. Additionally it is intriguing for its proposal of a new kind of mobile genetic element, cf-PISPs. The data are rich and the reasoning largely lucid, detailed, and sound. I would recommend this manuscript for publication.

We thank Reviewer #2 for their positive assessment of the study and for their insightful comments regarding the ecological interpretation of oyster-associated persistence and genomic diversity. Their suggestions helped us clarify the scope of the abundance analyses, improve figure readability, and more clearly articulate the role of oysters as reservoirs and sites of genetic exchange. We appreciate their supportive and thoughtful feedback.

2.1 Line 205: the idea that the environment inside oysters promotes *Vibrio*-phage coexistence (separate from its promotion of genomic plasticity and genetic exchange) is exciting, however the reasoning would be stronger if the authors could show clade-level bacteria-phage abundance was more correlated in oysters as compared to in seawater, i.e. Fig. 3B but for seawater as well. Alternatively, perhaps I mistook the authors' intent, in which case that might be made more clear.

We thank the reviewer for this comment and agree that the original wording could be misleading. Our intent in this section was not to test ecological correlations between phage and bacterial abundances in oysters versus seawater, but rather to examine whether the contrasting habitats experienced by *V. crassostreae* are reflected in patterns of genomic diversity and mobile genetic element content.

We have therefore revised the introduction of this section to clarify this objective. The revised text now explicitly frames the analysis as a genomic investigation of host populations sampled in parallel with phages during the same time series, rather than as a test of phage–host abundance correlations. We believe this clarification resolves the ambiguity noted by the reviewer.

2.2 Fig. 3B: it would be good to see the CI/SD/distribution of data as ribbons or intervals (cf. Fig. S12A), as it is difficult to tell anything about the shape of the distribution from the dense individual dots.

The Figure has been modified according to Reviewer 2 and 3 requests

2.3 Fig. 5A: 'no terminal repeats' are represented as diamonds in the figure, but the caption says triangles.

Figure 5A (temperate phage) has been changed by VIRIDIC intergenomic similarity as in Figure 2A for the lytic phages. we believe that this ease the comparison of the two data set.

2.4 Fig. S1: annotate XY axes to make clear which one is hosts and which phages.

The Figure S1 has been improved and is now Fig2. A which shows the intergenomic clustering (VIRIDIC), with phage genera annotated by the *V. crassostreae* clade used for isolation. Importantly, host clade information is not used in the clustering itself, but only to visualize associations.

Reviewer #3 (Remarks to the Author):

Overall assessment

This is a rich, carefully executed study that combines long-term field sampling, dense culture-based phage isolation, genomics, and MGE analyses to address how phage–host lineages behave over years in an open marine system. The dataset is genuinely impressive and many of the conclusions about lineage persistence, modular infection structure, and the diversity of MGEs in oyster-associated *Vibrio crassostreae* are well supported. I am enthusiastic about publication after revision.

My main reservations concern (i) the repeated framing of “ecological constraints” and “constraints on antagonistic coevolution” as the primary explanation for phage genomic stability, and (ii) the assumption that lytic phages are engaged in ongoing arms races rather than spending much of their time in a low-replication, persistence mode. These issues are conceptual rather than technical, and can largely be addressed by tempering language and more fully considering alternative explanations that parsimonious and align with the data presented than the “constraints” explanation does.

We are particularly grateful to Reviewer #3 for their careful conceptual analysis and constructive critique of the manuscript’s framing. Their comments prompted substantial improvements in how we interpret phage genomic stability, replication dynamics, and the role of ecological context. By encouraging us to explicitly contrast alternative, parsimonious scenarios and to temper causal claims, Reviewer #3 helped us sharpen the manuscript’s conceptual rigor and broaden its relevance for future work in microbial ecology and evolution.

Comments

3.1. “Ecological constraints” vs “limited replication / passive persistence”

The title, abstract, and Discussion repeatedly assert that “nested ecological constraints” or “environmental constraints” stabilize viral lineages and constrain evolution (e.g., Abstract lines 38–42; Discussion lines 400–417).

However, the primary observations are:

- o For several virulent species, there are phage genomes that are identical across four years (e.g., LS29), and others that differ by only a handful of SNPs.
- o For many temperate species, there is even less change, including prophages identical across two decades.

These patterns are very naturally explained by low cumulative replication and/or long-lived particles, rather than by strong constraints acting on rapidly evolving, frequently replicating populations. If the lytic phages were engaged in continuous arms races but constrained, one would expect:

- o measurable substitution rates with at least synonymous changes accumulating genome-wide, and
- o non-synonymous changes concentrated in interaction loci.

Instead, for some lineages there are no mutations at all, which seems more consistent with very few replication rounds between sampling points than with strong purifying selection under active coevolution.

Suggestions:

- o Please soften or rephrase the claim that ecological constraints per se are responsible for genomic stability. For example, the title could shift from “Ecological constraints foster...” to “Ecological context and low replication facilitate...” or similar.
- o In the Discussion, explicitly contrast two scenarios:
 1. Active phage–host arms races whose outcomes are tightly constrained; vs.
 2. Phage populations that spend most of the time in a low-replication persistence state (e.g., in sediments, on particles, or in low-MOI infections), with occasional bursts during host blooms.
- o Make clear that your data cannot distinguish between these mechanisms at present, and that passive persistence is a parsimonious explanation for the complete absence of sequence change in some lineages.

We thank the reviewer for this thoughtful and important critique. We agree that our initial framing may have overstated the role of “ecological constraints” acting on rapidly replicating phage populations, whereas the primary observations—complete absence of substitutions in some virulent phage lineages over four years, minimal divergence in others, and near-complete conservation of temperate phages over decades—are naturally compatible with scenarios involving low cumulative replication and/or long-lived viral particles.

In response, we have substantially revised the title, abstract, and Discussion to soften causal claims about ecological constraints per se and to explicitly consider alternative, parsimonious explanations. In the revised Discussion, we now put forward the two non-mutually exclusive scenarios: (i) phage populations that persist predominantly in a low-replication state, with virulent phages maintained as long-lived particles and replicating episodically during seasonal host blooms, and temperate phages persisting via vertical transmission as prophages; and (ii) recurrent phage replication that is nevertheless strongly structured by ecological and genetic context, including seasonal host availability, spatial compartmentalization within oysters, stable host clade structure, receptor specificity, and host defense systems encoded on mobile genetic elements.

We now explicitly state that our data do not allow us to distinguish conclusively between these mechanisms, and that passive persistence with limited cumulative replication is a parsimonious explanation for the complete absence of sequence change observed in some lineages. Rather than invoking a single dominant process, we emphasize that long-term phage genomic stability in this system likely emerges from the interplay between episodic replication, environmental persistence, intermittent antagonistic interactions, and host-associated ecological structure. These revisions clarify the scope of our conclusions and align the interpretation more closely with the patterns supported by the data.

3.2. Assumption of arms-race coevolution for virulent phages

Throughout, lytic phages are described as being in antagonistic coevolution or “arms races” with their hosts. Yet several lines of evidence argue against strong ongoing arms races in this specific system:

- o Abundances of phages and hosts are often positively correlated within oysters and across time; there is no clear predator–prey lag or signature of recurrent selective sweeps.

- o For some virulent species, you find no substitutions across years, rather than the accumulation of synonymous (and occasionally non-synonymous) changes expected under constrained arms-race dynamics.

- o You show experimentally that virulent phages retain viability after years of storage with only ~1 log drop in titer, suggesting that long-lived particles might also be possible in the environment.

Taken together, these observations look more compatible with phage persistence with limited replication than with continuous arms races.

We thank the reviewer for raising this important conceptual point. We agree that our original wording may have overstated the assumption of continuous arms-race dynamics for virulent phages in this system. We have therefore revised the manuscript to clarify that, while antagonistic coevolution between *V. crassostreae* and its phages clearly occurs, it is unlikely to operate as a constant, genome-wide arms race.

Previous work in this system has demonstrated active coevolution through the gain and loss of host defense systems and phage counter-defenses encoded on mobile genetic elements (Piel et al., 2022), establishing that antagonistic interactions do shape phage–host compatibility. However, our time-series data indicate that these interactions are likely intermittent and spatially structured, rather than continuously driving rapid evolutionary turnover. In particular, the absence of predator-prey lags, the positive co-occurrence of phages and hosts within oysters, and the lack of sequence divergence in some virulent phage lineages are more consistent with episodic replication and persistence phases than with ongoing selective sweeps.

Accordingly, we have tempered references to arms-race dynamics throughout the manuscript and now emphasize that phage-host coevolution in this system likely proceeds through bursts of antagonistic interaction embedded within broader ecological and life-history constraints, rather than through continuous reciprocal adaptation.

3.3. dn/ds (or related) analyses could help distinguish hypotheses

In Figure 5B you compare substitution rates of virulent vs temperate phages, but the manuscript does not assess the distribution of non-synonymous vs synonymous substitutions.

- o If ecological or functional constraints are indeed limiting evolutionary outcomes despite ongoing mutation, one would expect $dn/ds < 1$ genome-wide, with especially low ratios in critical structural and receptor-binding genes.
- o Conversely, if there is very little replication, dn and ds are both near zero and the ratio is difficult to interpret.

Suggestions:

- o For lineages with sufficient polymorphism (e.g., LS32, perhaps some prophage species), please compute dn/ds or an analogous metric (e.g., pN/pS) at least at the whole-genome level, and ideally partitioned into structural vs non-structural genes.
- o Explicitly discuss how these results bear on the “constraints vs persistence” question.
- o In spots where you currently invoke constraint based purely on low substitution rates (e.g., Discussion lines 411–417), consider replacing those claims with interpretations supported by dn/ds , or softening if the data are inconclusive.

We thank the reviewer for suggesting these analyses to strengthen our interpretation of long-term persistence in the viral genomes. We now have included the results of dN/dS analyses as Figure S31 including comparisons between persistent lytic phage and prophage species and between structural and non-structural genes. We show an overall imprint of purifying selection as the median dN/dS is in all cases significantly below 1, but do not detect significant differences in either per-gene dN/dS or in the percentage of codons under selection between structural and non-structural genes.

3.4. Interpretation of “unexpected ecological forces”

Related to comments 1–3, some parts of the Discussion imply that there must be highly unexpected ecological forces maintaining stability (e.g., nested ecological filters constraining diversification).

In fact, the idea that phage particles can persist for long times with minimal replication—and that phage–host contact may be intermittent—is not particularly exotic and is consistent with your own laboratory stability data. The danger is that readers may leave thinking they should look for special “constraint mechanisms” rather than for mechanisms that permit viable phage persistence for years (e.g., low decay in cold sediments, adsorption to particulate matter or biofilms, repeated “reseeding” from refugia).

Suggestion:

Reframe these sections to emphasize that:

- o Your data highlight the surprising longevity of some virulent phage lineages;
- o The main mechanistic question for future work is how phages remain viable for years with low apparent replication, not necessarily what exotic ecological constraints suppress evolution during an arms race.

This is an important distinction because it will shape how follow-up studies are designed.

We thank the reviewer for this important clarification. We agree that our initial phrasing may have unintentionally suggested the existence of unusual or exotic ecological forces actively constraining phage evolution. We have revised the Discussion to reframe this interpretation and to emphasize that our primary observation is the unexpected longevity of some virulent phage lineages, rather than the action of specific constraint mechanisms.

In the revised manuscript, we now highlight that long-term phage persistence can plausibly arise from relatively simple and well-established processes, including low cumulative

replication, slow decay of virions, intermittent phage–host contact, and episodic replication during seasonal host blooms. These interpretations are consistent with our laboratory data showing long-term virion viability and do not require invoking strong or unusual selective constraints acting on rapidly evolving populations.

We further clarify that the key mechanistic question raised by our study is not necessarily what suppresses evolution during continuous arms races, but rather how phages remain viable and ecologically relevant over multi-year timescales despite limited apparent replication. We agree that this distinction is critical for guiding future work and explicitly discuss directions for follow-up studies, including identifying environmental reservoirs, quantifying decay rates, and assessing spatial refugia for phages in natural systems.

3.5. Contamination controls for identical genomes across years

The conclusion that identical phages were recovered four years apart is central, but the methods do not explicitly describe contamination controls between the 2017 isolates and the 2021 sampling campaign. The Materials and Methods section explains how plaques were purified and how high-titer stocks were stored, and notes that 2017 stocks remained viable after four years.

Given that:

- o 2017 phage stocks were stored in the same lab and remained viable; and
- o 2021 phage isolations, amplifications, and sequencing appear to have been done in the same general setting,

there is a non-trivial risk that a stored stock could have been inadvertently introduced into 2021 plates (e.g., via aerosols, shared pipettes, or reuse of phage stocks as positive controls).

Suggestions:

- o Explicitly state how cross-contamination between 2017 and 2021 isolates was prevented (physical separation of stocks and workspaces, separate freezers, no use of archived phages as positive controls, etc.).
- o If available, mention any negative controls that were processed in parallel (e.g., mock overlays with buffer) and their results.
- o If contamination cannot be definitively ruled out, acknowledge this limitation when claiming “identical genomes 4 years apart,” even if you regard it as unlikely.

We thank the reviewer for raising this important point. We agree that the possibility of cross-contamination between sampling campaigns must be explicitly addressed given the centrality of the observation that some phage genomes are identical across years. We have therefore revised the Materials and Methods to acknowledge this potential limitation and to clarify the measures taken to minimize cross-contamination.

Specifically, phage isolations and high-titer stock preparations for the 2021 campaign were conducted without handling archived 2017 phage stocks, which were stored separately, and a dedicated refrigerator and freezer were reserved for the 2021 collection. Archived phages were not used as positive controls during the 2021 isolation and amplification procedures. Although the scale of the screening effort (153 host strains across 35 sampling dates) precluded the systematic inclusion of mock controls, all phages were purified through up to three successive rounds of plaque isolation, substantially reducing the likelihood of carryover contamination.

We further note that, while accidental contamination cannot be formally excluded, the recovery of closely related but non-identical phage genomes across years is inconsistent with simple laboratory carryover and supports the biological origin of the observed persistence patterns. We have tempered our wording accordingly and explicitly acknowledge this limitation in the revised manuscript.

3.6. Claims about unprecedented scale

The manuscript describes the dataset as “unprecedented” in scale (e.g., Abstract lines 31–32, Results lines 76–82).

While the combination of >1,000 phages and 600 host genomes for a single species is indeed impressive, prior work (notably from Polz and collaborators) has produced very fine-scale, high-frequency time series of marine phage–host dynamics, in some cases with denser temporal sampling.

Suggestion:

Soften the language to something like “a large, well-resolved dataset” or “a uniquely dense dataset for a single bacterial species–phage assemblage,” and explicitly acknowledge prior high-resolution time-series work (e.g., Polz lab) as complementary.

We thank the reviewer for this comment and agree that the term “unprecedented” was too strong. We have softened the language throughout the manuscript, replacing it with formulations such as “a uniquely dense dataset for a single bacterial species–phage assemblage.” We now explicitly acknowledge prior high-resolution time-series studies of marine phage–host dynamics, notably work from the Polz laboratory, and position our dataset as complementary. This includes previous collaborative studies between our teams (e.g., Hussain et al., *Science*; Piel et al., *Nature Microbiology*), which documented fine-scale coevolutionary dynamics in marine *Vibrio* populations. While earlier studies achieved very fine temporal resolution, our present work provides dense culture-based sampling, complete genomes, and host–phage resolution across multiple years for a single bacterial species in an animal-associated environment.

3.7. Connection to experimental work showing long-lived lytic lineages (Leap-Frog dynamics)

You note that experimental coevolution can produce rapid diversification (e.g., for *Synechococcus* cyanophages), but there is also experimental literature where lytic phage lineages persist cryptically for long periods despite the potential for rapid evolution (e.g., Leap-Frog dynamics experiments, where “hidden” lineages re-emerge after hundreds of generations).

Suggestion:

o In the Introduction or Discussion (perhaps around lines 50–57 or 380–387), cite the Leap-Frog Dynamics work and point out that your field observations of long-lived virulent lineages are conceptually similar, and could potentially be reproduced in controlled lab systems.

We thank the reviewer for pointing out this important connection. We agree that experimental work on “leapfrog” dynamics provides a valuable conceptual framework for interpreting the persistence of virulent phage lineages observed in our field data. We have therefore added a reference to the leapfrog dynamics literature (Gupta et al., 2022) in the revised Discussion, and explicitly note that the long-term persistence of some lytic phage lineages in our system is conceptually consistent with experimental observations in which cryptic viral lineages persist for extended periods and re-emerge despite the potential for rapid evolution.

3.8. Figure 3B readability

The x-axis tick labels in Figure 3B are very hard to read in the current form. Please increase font size or reduce the number of ticks.

The x-axis ticks have been simplified.

3.9. Host range bias from using archival strains as bait

You use a panel of 153 archival *V. crassostreae* strains as bait for phage isolation in 2021. This is sensible for consistency, but it implicitly biases detection toward phages that infect those specific strains.

Please note this as a limitation: phages specialized on 2021-specific lineages that are poorly represented in the archival collection might be under-sampled.

We agree that using a fixed panel of archival *V. crassostreae* strains as bait for phage isolation constrains the host range of detectable phages. This choice was intentional and reflects the primary goal of our study, which was not to exhaustively sample phage diversity in 2021, but to enable direct comparisons across years at a consistent ecological and genetic scale.

By using the same host clades and strains previously characterized, we ensured that differences in phage isolation reflected changes in ecological dynamics rather than shifts in host sampling. Our analyses focus on the repeated recovery of the same phage genera associated with the same bacterial clades, allowing us to track persistence and temporal dynamics of these host–phage lineage pairs. Importantly, we do not reconstruct cross-infection matrices nor infer host range breadth from these data.

While phages specialized on rare or newly emerging host lineages may therefore be under-sampled, this does not affect our main conclusions regarding lineage persistence, clade–genus associations, or the ecological and genomic patterns reported here.

Recommendation

In summary, I support publication of this manuscript after revision. The work presents impressive and novel data, and the conceptual advances around phage and MGE diversity in an oyster-associated system are substantial. Addressing the points above will make the paper both more rigorous and more useful to researchers who build on this system.

Answer to the REVIEWER 1 COMMENT

Reviewer #1 (Remarks to the Author):

This was a strong paper before and stronger now with appropriate controls and caveats. The authors have done an exceptional job in taking reviewer comments and addressing issues, correcting claims when necessary, and writing a paper that is easier to interpret and will be more durable. I commend them. I could quibble here and there, but won't. Instead, I make one suggestion.

On page 13, the authors interpret their findings by writing: "Two non-mutually exclusive scenarios may account for the observed genomic stability of some phage lineages." I am reminded of a column "In theory" by the Nobelist Sydney Brenner in Current Biology (can be found here <https://tavernarakislab.gr/news/Loose-Ends-and-False-Starts.pdf>), in which Brenner writes:

I recall a meeting in the 1970s where a speaker presented two different models of transposition, which we can call A and B. The climax of the talk came when the speaker triumphantly declared that there were only two possibilities: "Either A is right and B is wrong, or B is right and A is wrong." He had to be reminded that he had overlooked a third possibility which was that they were both wrong.

I would simply caution that given the complexity of the data, the two scenarios the authors envision are not the only ones and it would be important to remind the reader that there may be new eco-evolutionary mechanisms at play. The finding of stability is precisely one of the reasons I think this paper will attract readers and new hypotheses.

We edited the sentence as "At least two non-mutually exclusive scenarios may account for the observed genomic stability of some phage lineages."

Reviewer #2 (Remarks to the Author):

Authors have addressed my comments on the original manuscript.

Reviewer #3 (Remarks to the Author):

I'm happy with the revisions and responses to my review. Because of time restrictions, I was unable to evaluate responses to the other reviewers.